

# Comparison of quite time ionospheric total electron content from IRI-2016 model and GPS observations

Mulugeta Melaku[1] and Gizaw Mengistu Tsidu[2]

[1]Department of Physics, Addis Ababa University, Addis Ababa, Ethiopia
[2]Department of Earth and Environmental Sciences, Botswana International University of Science and Technology, Palapye, Botswana

**Correspondence:** Gizaw Mengistu Tsidu (mengistug@biust.ac.bw)

**Abstract.** Earth's ionosphere is an important medium of radio wave propagation in modern times. However, the effective use of ionosphere depends on the understanding of its spatio-temporal variability. Towards this end, a number of ground and space-based monitoring facilities have been set up over the years. This is also complemented by model-based studies. However, assessment of the performance of the ionospheric models in capturing observations needs to be conducted. In this

work, the performance of IRI-2016 model in simulating total electron content (TEC) observed by network of global position System (GPS) is evaluated based on RMSE, bias, correlation and categorical metrics such as Quantile Probability of Detection (QPOD), Quantile False Alarm Ratio (QFAR), Quantile Categorical Miss (QCM), and Quantile Critical Success Index(QCSI). IRI-2016 model simulations are evaluated against GPS-TEC observations during the solar minima 2008 and maxima 2013. Higher correlation, low RMSE and bias between the modeled and measured TEC values are observed during solar minima

than solar maxima. The IRI-2016 model TEC agrees with GPS-TEC strongly over higher latitudes than over tropics in general and EIA crest regions in particular as demonstrated by low RMSE and bias. However, the phases of modeled and simulated TEC agree strongly over the rest of the globe with the exception of the polar regions as indicated by high correlation during all solar activities. Moreover, the performance of the model in capturing extreme values over magnetic equator, mid- and high-latitudes is poor. This has been noted from a decrease in QPOD, QCSI and an increase in QCM and QFAR over most of the

globe with an increase in the threshold percentile values of TEC to be simulated from 10% to 90% during both solar minimum and maximum periods. The performance of IRI-2016 in correctly simulating observed low (as low as $10^{th}$ percentile) and high (high than $90^{th}$ percentile) TEC over EIA crest regions is reasonably good given that IRI-2016 is a climatological model despite large RMSE and positive model bias. Therefore, this study reveals the strength of the IRI-2016 model, which was concealed due to large RMSE and positive bias, in correctly simulating the observed TEC distribution during all seasons and

solar activities for the first time. However, it is also worth noting that the performance of IRI-2016 model is relatively poor in 2013 compared to that of 2008 at the higher ends of the TEC distribution.

## 1   Introduction

Radio wave has turned into indispensable and spectacular means in the progress of space satellite communication and navigation. Earth's ionosphere is an essential medium for the propagation of radio wave signals (Mengistu and Abraha, 2014,
and references therein). Kumar et al. (2014) noted that dual-frequency global position system (GPS) receivers have increased over the whole world with numerous networks in order to enhance applications based on space satellite communications and navigations. However GPS signals are influenced by ionospheric disturbances due to the existence of electron. Total Electron Content (TEC) is a parameter which provides full picture of the ionosphere. It is an essential ionospheric property for the

investigation of ionospheric variability and dynamics since TEC changes as a function of geographic location, time of the day, day of the season, season of the year, and solar and geomagnetic activities. As a result, several authors have investigated the distribution and characteristics of TEC variations (Mukherjee Shweta, 2010; Sethi et al., 2011; Mengistu and Abraha, 2014; Bardhan et al., 2014; Saranya et al., 2014; Grynyshyna-Poliuga et al., 2015; Themens and Jayachandran, 2016; Sharma et al., 2017; Venkata Ratnam et al., 2017; Perna et al., 2017; Rao et al., 2018; Perna et al., 2018). Moreover, the coupling of lower

atmosphere and ionosphere as well as coupling of thermosphere and magnetosphere contribute to the complexity of the TEC variability in the ionosphere (Oberheide and Gusev, 2002; Takahashi et al., 2005, 2006, 2007, 2009; Luhr et al., 2007; Wu et al., 2009; Liu et al., 2010; Onohara Batista and Takahashi, 2013; Mengistu and Abraha, 2014). Understanding TEC variability allows us to characterize time delay in radio wave signals induced by ionosphere as the delay is directly proportional to TEC and develop ionospheric model to predict TEC.

Historically, as early part of efforts to understand the Earth's upper ionosphere, the first satellites radio-measurement recorded some crucial results of the upper atmosphere. Measurements of Orbital period through radio-locations revealed that temperature and density show high discrepancies in the upper atmosphere (Bilitza et al., 1990). Hence, decision has been taken by the formerly established Committee on Space Research (COSPAR) to develop a set of empirically based tables expressing these

new results. Thus, the new results are presented under the name of COSPAR International Reference Atmosphere (CIRA) since 1961. A few years later after the achievement of CIRA, S. Bowhill proposed a reference to be named as International Reference Ionosphere (IRI) to represent the ionized constituents of the atmosphere (Rawer et al., 1978; Bilitza , 1986; Rawer, 1988; Bilitza et al., 1990). The International Union of Radio Science (URSI) begun to cooperate with COSPAR on the IRI since 1969. Thus IRI models are managed by COSPAR and URSI.

IRI, one of the empirical modeling tools currently available to a wider scientific community, portrays spatial and temporal monthly mean of the ionosphere, for a specific solar variability (Bilitza et al., 1990; Bilitza, 2001; Bilitza et al., 2014). IRI may forecast monthly TEC variability better than TEC daily variability since IRI model is a climatological model and its parameters are derived based on the availability of ground, in-situ and space-based measurements. Therefore, IRI is globally

recognized as the guideline for ionospheric parameters and is applied by various scholars in comparison studies with TEC derived from GNSS GPS networks and in other studies (Kouris and Fotiadis, 2002; Kouris et al., 2004; Zhang et al., 2006; Mukherjee Shweta, 2010; Sethi et al., 2011; Bilitza et al., 2011; Bardhan et al., 2014; Asmare et al., 2014; Saranya et al., 2014; Grynyshyna-Poliuga et al., 2015; Patel et al., 2015; Mengistu et al., 2016; Wang et al., 2016; Themens and Jayachandran, 2016; Perna et al., 2017; Venkata Ratnam et al., 2017; Sharma et al., 2017; Perna et al., 2018; Rao et al., 2018). For instance, in the

older versions ( IRI-2000, IRI-2001), it is found that IRI is overestimating both ionogram and GPS TEC values (Mosert et al.,





2007). Praveen et al. (2010) found the estimations of IRI-2007 model have seasonal and longitudinal discrepancies in TEC over low-latitude stations. Kenpankho et al. (2011) have also noted that IRI-2007 underestimates GPS TEC over an equatorial region in Thailand with poorer performance during day than night times. Venkata Ratnam et al. (2017) have found that IRI-2007 and IRI-2012 models capture observed GPS-TEC at two low latitude GPS stations in India except during dawn hours

(01:00-06:00 LT) when the models overestimate TEC. The authors have also revealed presence of higher percentage deviations during equinoctial months than summer. Moreover, the authors noted limited skill of the models in capturing observed TEC changes during June storm 2013 although there is some difference between the two versions of the model. In fact, the poor skill of IRI in simulating TEC during geomagnetic storms has been reported by numerous other authors as well (e.g., Asmare et al., 2014; Tariku, 2015). The weakness of IRI-2012 is not only limited to storm events. Kumar (2016) determined that performance

of IRI-2012 in simulating TEC over the global equatorial region is better during a deep solar minimum (2009) than a maximum year (2012) as the IRI-2012 model overestimates the observed GPS-TEC at all equatorial stations with larger discrepancy from observations during solar maximum (2012) than during solar minimum(2009). The author also noted difference between seasons with the maximum discrepancy during the December solstice and minimum during the March equinox. The IRI model has gradually been revised methodically to address many of these limitations in subsequent versions that led to improvement in

its forecasting skill in the course of several upgrades to the latest version: International Reference Ionosphere-2016 (IRI-2016) (Bilitza et al., 1990, 1993a, b, 2014, 2017).

Important advancement of IRI-2016 model version has been made based on ground and space-based observations (e.g., ionosonde and radio occulation). The major changes include two new model options for the F2 peak height hmF2, revised

solar indices and an improved modeling of topside ion densities at wider range of solar activities. The major amendment in the IRI-2016 is the inclusion of AMTB2013 (ionosonde-based) and SHU-2015 (GNSS radio occulation-based) F-peak height hmF2 models. As further improvement requires identification of the weakness and strength of this latest version, the assessment of the performance of IRI-2016 has been ongoing. Early results at selected locations have shown some improvements over its predecessors. For example, Mengistu et al. (2018) have shown that IRI-2016 performed better than NeQuick and IRI-2012

in estimating monthly mean TEC observed by three of the four ground-GPS receivers in Ethiopia. On the other hand, Rao et al. (2018) have shown that existence of significant discrepancy between IRI-2016 and ionosonde observations of foF2 over China EIA crest region during different seasons and local times for 2008–2013 period. These contrasting results necessitate comprehensive evaluation of the model globally at different spatio-temporal scales. Moreover remotely sensed data models and satellite measurements are exposed to biases and uncertainties emerging from environmental and mathematical aspects.

Common metrics for validation of model generated TEC involves evaluation of RMSE, bias and correlation. However, most of the contribution to bias and RMSE usually comes from the extreme ends of the TEC distribution during a given month, season or solar year. It is customary to investigate such contribution using scatter plot qualitatively. Recently, quantile based categorical metrics such as quantile probability of detection, quantile critical success index etc have been proven to be important tools to assess these biases and scatter at the extreme ends of data distribution quantitatively as demonstrated in other

disciplines (e.g., AghaKouchak et al., 2009; Gilleland et al., 2009; Entekhabi et al., 2010; Dorigo et al., 2010; Gebremichael,



2010; AghaKouchak et al., 2011).

Therefore, this paper focus on the comprehensive global validation of IRI-2016 model on monthly, seasonal and annual TEC variations based on observed TEC by network of ground-GPS receivers run by International Global Navigation Satellite System(GNSS) Service (IGS) using the common statistical metrics and the quantile-based categorical metrics. To our knowledge,

there is no comprehensive and global evaluation of IRI-2016 that includes detailed analysis at the tails of TEC distribution. The paper is organized such that Section 2 highlights data and methodologies employed for validation of IRI-2016 TEC data against IGS GPS TEC. Section 3 covers results and discussion while Section 4 provides conclusion.

## 2    Data and methodology

### 2.1    Data

#### 2.1.1    GPS

The TEC data extracted at a grid resolution of $5^0$ latitude by $5^0$ longitude from IGS, hereafter referred to as GPS-TEC for solar minimum 2008 and solar maximum 2013 are used. The data is processed using 422 ground-based GPS receivers (see Fig. 1 and Table 1) from 32 GPS satellites for the solar minima 2008 and solar maxima 2013 as shown in Table 1. TEC is measured by GPS signals through integration of the electron density profile. The differential phase, $\Delta\Phi$, of the two waves on L1 and L2

bands of dual frequency GPS can be used to determine TEC according to procedure described by many authors (Bossler et al., 1980; Melbourne et al., 1994; Morgan and Johnston, 1995; Axelrad et al., 1996; Komjathy, 1997; Schreiner et al., 1999; Parssinen et al., 1999; Hajj et al., 2000; Woo, 2000; Hajj et al., 2002; Oloufa et al., 2003; Borghetti et al., 2006; Hoffmann and Jacobi, 2006; Hernandez et al., 2011; Pradhananga and Teizer, 2013).

GPS data is filtered using Dst data such that days with geomagnetic storms are excluded from the comparison since it has been

indicated in several other studies that IRI models are insensitive to the storm option and fails to reproduce observed TEC on storm days (e.g., Asmare et al., 2014; Tariku, 2015).

**Table 1.** GPS stations and satellites across the world.

| Code | Ground-GPS Stations | Satellites |
|------|---------------------|------------|
| IGS  | 422                 | 32         |

#### 2.1.2    TEC from IRI-2016 model

TEC data is simulated using IRI-2016 as function of universal time and geographical grids that matches the spatio-temporal grids of observed IGS GPS-TEC for the two selected years. The model is configured such that the URSI and NeQuick2

options for F-peak model and for the top-side profile estimation have been considered in this study. Furthermore, the newly added Shubin-Cosmic model for hmF2 and ABT-2009 option for the bottom side thickness shape parameter are considered.





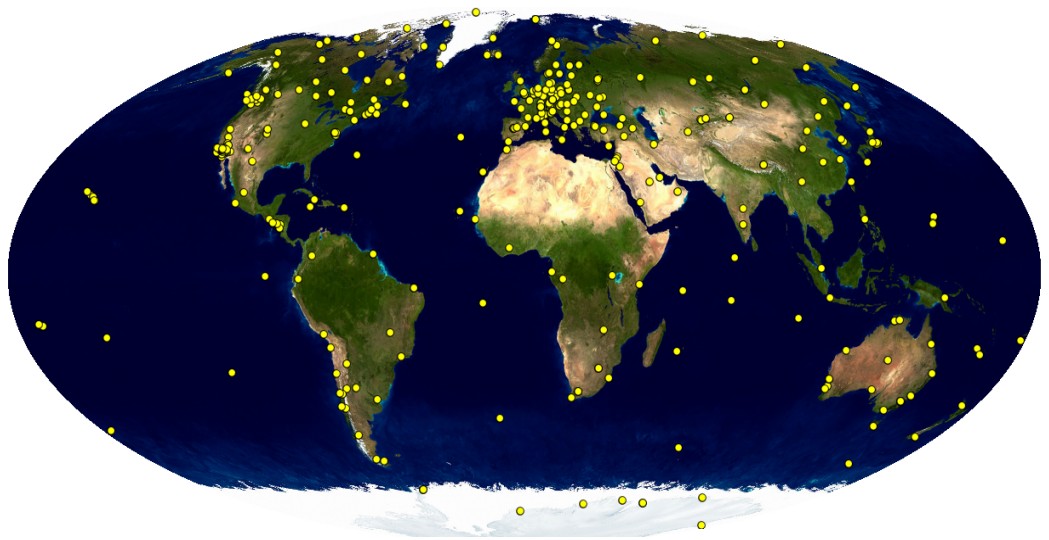

**Figure 1.** Distribution of the GPS station from IGS Network

(Source: http://www.citg.tudelft.nl/fileadmin/Faculteit/LR/Organisatie/Afdelingen...).

Moreover, the storm related models were set to off. The Shubin-Cosmic model was developed with a large amount of radio occulation (RO) data from CHAMP, GRACE and COSMIC and with hmF2 data from 62 digisondes for the years 1987-2012 from the Digital Ionogram Data Base [http://ulcar.uml.edu/DIDBase/] (Shubin et al., 2013; Shubin, 2015; Bilitza et al., 2017, and references therein). Moreover, the historical development of IRI and details of the recent IRI-2016 model are given by
5   Bilitza et al. (2017).

### 2.1.3   Disturbance storm time (Dst)

The Dst index represents the axially symmetric disturbance magnetic field at the dipole equator on the Earth's surface. Major disturbances in Dst are negative, namely decreases in the geomagnetic field. Therefore, days with Dst greater than -30 nT are assumed to be quite days and therefore included in the comparison of IRI-2016 and GPS-TECs. The two-hourly Dst data is
10   obtained from http://www.wdc.kugi.kyoto-u.ac.jp/dstdir/ for the two years.





## 2.2 Methodology

### 2.2.1 Numerical Statistics: RMSE, Bias and Correlation

Monthly comparison of TEC from IRI-2016 Model and GPS measurements are evaluated with root mean square error (RMSE),
systematic error (Bias) and pattern correlation between them for the selected years. RMSE, which is the square root of the mean
of all errors, indicates the deviation between simulated and observed data. It is given as

$$RMSE = \sqrt{\frac{1}{n}\sum_{i=1}^{n}(S_i - O_i)^2}, \tag{1}$$

or

$$RMSE = \sqrt{\sigma_S^2 + \sigma_O^2 - 2\sigma_S\sigma_O R + (Bias)^2}. \tag{2}$$

in terms of individual standard deviations (variances) of the simulations ($\sigma_S$), observations ($\sigma_O$) as well as bias and correla-
tion (R) between the two data sets: $\sigma_S^2 = \frac{1}{n}\sum_{i=1}^{n}(S_i - \bar{S})$, $\sigma_O^2 = \frac{1}{n}\sum_{i=1}^{n}(O_i - \bar{O})$ and $R = \frac{1}{n}\sum_{i=1}^{n}(S_i - \bar{S})(O_i - \bar{O})/\sigma_S.\sigma_O$.
The systematic error (Bias) discloses the mean difference between the simulated (IRI-TEC) and measured (GPS-TEC) data:

$$Bias = \frac{1}{n}\sum_{i=1}^{n}(S_i - O_i) \tag{3}$$

where $S_i$ and $O_i$ are simulated and observed total electron content values respectively and n is the total number of data
points for comparison.
The correlation coefficient is derived from the covariance of the simulated and observed variables divided by the product of
their standard deviations. Moreover the correlation coefficient (R) is an implication of how much both the spatial and temporal
patterns in the IRI-2016 Model match the IGS-GPS observations (Murphy, 1998; Taylor, 2001; Daniel., 2006; Ochoa et al.,
2014):

$$R = \frac{cov(S,O)}{\sigma_S\sigma_O} = \frac{\frac{1}{n}\sum_{i=1}^{n}(S_i - \bar{S})(O_i - \bar{O})}{\sqrt{\frac{1}{n}\sum_{i=1}^{n}(S_i - \bar{S})^2}\sqrt{\frac{1}{n}\sum_{i=1}^{n}(O_i - \bar{O})^2}} \tag{4}$$

### 2.2.2 Categorical Statistics: Quantile Probability of Detection (QPOD), Quantile False Alarm Ratio (QFAR),
Quantile Categorical Miss (QCM) ,Quantile Critical Success Index (QCSI)

Categorical statistics employed in this study aim to evaluate the extent to which the simulation captures the distribution of the
observed GPS-TEC above certain selected thresholds. As IRI model is empirical model based mainly on past observations, it is
natural to expect that its performance at the extreme ends of the observed distribution may suffer from inaccuracies. However,
the extent of this discrepancy at the extreme ends of the observed TEC distribution is not fully assessed. Therefore, categorical





statistics such as QPOD, QFAR, QCM, and QCSI are employed to evaluate the performance of IRI-2016 model in simulating the whole spectrum of observed distributions from low to high extreme TECs. The QPOD defines part of the observations (OBS) above selected percentile threshold (t) identified accurately by the simulation (SIM). It is given by

$$QPOD = \frac{H}{(H+M)} \tag{5}$$

where H and M stand for hit and miss rates respectively. H and M are given in terms of t, $OBS_i$ and $SIM_i$ as follows: H = $\sum_{i=1}^{n}(SIM_i|(SIM_i > t \ \& \ OBS_i > t))$ and M = $\sum_{i=1}^{n}(OBS_i|(SIM_i \leq t \ \& \ OBS_i > t))$. A perfect detection signifies that the miss rate is zero implying that QPOD equals 1. In contrast a model with no skill has zero hit rate which suggests a QPOD value of zero. Therefore, QPOD attains a value of 0 for no skill and 1 for perfect score (Behrangi et al., 2011; AghaKouchak and Mehran, 2013).

The QFAR quantifies TEC above the selected threshold detected by simulation but not available in observations. The QFAR covers from 0 to 1; 0 signify perfect score (Brown et al., 2004; AghaKouchak and Mehran, 2013):

$$QFAR = \frac{F}{(H+F)} \tag{6}$$

where F stands for false alarm rate and is given as

F = $\sum_{i=1}^{n}(SIM_i|(SIM_i > t \ \& \ OBS_i \leq t))$.

The QCM may be defined as 1 - POD which ranges from 0 to 1, with 0 being the perfect score. QCM can be given specifically in terms of hit and miss rate as:

$$QCM = 1 - POD = 1 - \frac{H}{(H+M)} = \frac{M}{(H+M)} \tag{7}$$

The QCSI combines various features of the QPOD and QFAR, to determine the total skill of the simulation relative to obser-

vation as a function of H, F and miss rate (M):

$$QCSI = \frac{H}{(H+M+F)} \tag{8}$$

The QCSI ranges from 0 (no skill) to 1 (perfect skill) (Davis et al., 2009; AghaKouchak and Mehran, 2013). For example, a QCSI of 0.7 indicates that the simulation detects 70% of observed TEC above certain percentiles.

Both of these numerical (continuous) and categorical statistics are used to assess the model skill in capturing the individual observations for each calendar months and seasons for the solar minimum and maximum years.





## 3 Results and Discussions

### 3.1 Numerical statistics: RMSE, Bias, R

#### 3.1.1 Comparison of IRI-2016 simulation and GPS-TEC observations on a monthly basis

Fig. 2 shows RMSE, Bias and R between IRI-2016 TEC simulations and GPS-TEC observations for all the 12 months during
solar minimum 2008 period. Fig. 3 shows also RMSE, Bias and correlation as Fig. 2 but for the solar maximum period 2013
using same color scale. The RMSE in Fig. 2 is in the range of 0.5 to 11.9 TECU during 2008. The range of RMSE in 2013
(0.5 to 23.3 TECU) is much higher than that of 2008. Moreover, the RMSE of IRI TEC with respect to GPS TEC during
2008 is within 0.5 to 4.3 TECU over most of the globe with the exception of tropics which exhibits RMSE in the range of 4.3
to 11.9 TECU. In contrast, in 2013, the RMSE outside tropics ranges from 0.5 to 11.9 TECU and it varies from 4.3 to 23.3
TECU over tropics in 2013 implying IRI-2016 model exhibited poor performance in capturing observed GPS-TEC over the
same region during the 2013 solar maximum period as demonstrated by very high RMSE (Fig. 3). The difference between the
model and GPS TEC over the EIA crest regions is much higher than the rest of the globe as noted from high RMSE during both
solar activity years (Figs. 2-3). Moreover, the summer hemispheres experience higher RMSE than winter hemisphere with high
(low) RMSE from May to August over northern (southern) hemisphere and vice verse from October to January during solar
maximum period of 2013 (Fig. 3). There is no similar apparent hemispheric differences with seasons during solar minimum
2008 period.

The IRI-2016 TEC is low biased up to -7 TECU over most of the globe with respect to GPS-TEC during 2008. A positive
bias is notable over EIA crest regions throughout the whole period in 2008. However, maximum positive bias in IRI TEC
with respect to GPS TEC is observed over EIA crest regions from August to December (Fig. 2). The bias along EIA crest
region shows longitudinal variation with most of the peaks located in African and American longitude sectors. The IRI TEC
over the Asian longitude sector is low biased along the EIA crest region during most of the months (Fig. 2). This is consistent
with previous investigation by Kenpankho et al. (2011) who found that IRI-2007 underestimates GPS-TEC with a maximum
difference of 15 TECU during day times and a minimum variation of 5 TECU during night times over an equatorial region in
Thailand. Grynyshyna-Poliuga et al. (2015) have also shown that the TEC derived from the IRI-2012 model over mid-latitude
station, Warsaw, was generally low biased with respect to the GPS-TEC. The maximum differences are about 10 TECU during
the daytime and 2 TECU during the nighttime. As noted by other authors (e.g., Akala et al., 2015, and references therein)
contribution from the plasmasphere above 2000 km in GPS-TEC might have contributed to the discrepancy. For example,
Akala et al. (2015) have found that the contribution of Plasmasphere electron content to GPS-TEC is maximum during the
December solstice and minimum during the June solstice. Moreover, the authors noted that plasmaspheric TEC contribution to
GPS-TEC is varying with respect to solar activity. The discrepancy between the IRI-2016 TEC simulations and GPS-TEC can
not be fully attributed to the plasmasphere TEC since positive biases in IRI-2016 TEC are evident along some longitude sectors
of EIA crest regions. This positive bias with maximum (up to 16.5 TECU) during day and minimum during night time was



also reported by Wan et al. (2017) at four stations in China covering the EIA crest region. However, this modest discrepancy is not the case in 2013 during solar maximum period when wide spread negative bias in IRI TEC of up to -18 TECU in May in the northern hemisphere, and during November and December in the southern hemisphere was observed. Moreover, negative bias was also prevalent over most of northern hemisphere from June to August and over most of southern hemisphere from

October to December and in January. In contrast the period from July to September is characterized by positive bias in IRI TEC over most of southern hemisphere (Fig. 3). The weak performance of IRI model during solar maximum is also reported by Venkatesh et al. (2014) who have conducted comparison of GPS-TEC with The IRI-2012 modeled TEC over Brazilian region during the periods from 2010 to 2013. Li et al. (2016) determined low bias in night-time model TEC during solar minimum (2009) and maximum (2013), and good (poor) agreement between periodic components of TEC-IRI in low (high) solar activity

year 2009 (2013) and TEC derived from Global Ionosphere Map (TEC-GIM) at the Beijing Fang Shan station. Moreover, the IRI-2016 model was investigated for its performance over four stations in Ethiopia within the equatorial latitudes by Mengistu et al. (2018). The authors revealed better performance during solar minimum (2008) than solar medium (2011) activity years. Other studies with focus on high latitudes have shown similar weakness in IRI model. For example, IRI is also shown to significantly underestimate the magnitude of solar cycle variations in TEC and underestimate monthly median TEC at high solar

activity by as much as 15 TECU (Themens and Jayachandran, 2016) which are greatest during the equinoxes and significant during summer periods but are lowest during winter median TEC. These asymmetries suggest that the IRI-2016 has weakness to capture enhanced TEC during summer of each hemisphere when the sun is overhead in each hemisphere at time of maximum solar activity. This is partly inherent in its nature as IRI is based on mean observations to develop empirical formulations.

In contrast to simulation of the actual magnitude of TEC (as demonstrated by high negative bias), IRI-2016 performs well in capturing the phase of variation of TEC irrespective of seasons and the nature of solar activity as demonstrated by high correlation with GPS-TEC during both periods (2008 and 2013) (see Figs. 2-3). However, IRI-2016 exhibits weak performance over high latitudes (low correlation of up to 0.17 or negative correlation in some months) as compared to tropics and mid-latitude with correlation as high as 0.98. In addition there is also evidence of poorer correlation between simulated IRI TEC

and observed GPS TEC over high latitudes in south-western hemisphere during solar maximum 2013 than solar minimum 2008 periods (see Figs. 2-3). Table 2 shows the global average performance of IRI-2016 with respect to GPS TEC observations in 2008 and 2013. The table also includes global maximum and minimum RMSE, bias and correlation. The lowest RMSE of 0.48 TECU and 0.74 TECU in July are observed during 2008 and 2013 respectively. Similarly, highest RMSE of 11.63 in October and 24.27 TECUs in November are determined during 2008 and 2013 respectively. The global average RMSE varies

from 2.16 TECU in July to 3.64 TECU in March during solar minimum period of 2008. In contrast, in 2013, the mean RMSE varies from lowest value of 4.78 TECU in July to highest value of 9.39 TECU in December indicating the model performance is poorer in 2013 than in 2008 with significant seasonal variation in the IRI-2016 model skill. The globally averaged bias varies from -2.23 TECU in April to -0.76 TECU in October in 2008. However, it varies from 0.04 TECU in September to -6.4 TECU in December in 2013 (see Table 2). The spatial mean of seasonal correlation varies from 0.78 in January and December to 0.91

in September in 2008 while it varies from 0.80 to 0.88 in 2013 through January to December.





**Table 2.** Statistical parameters for monthly comparisons of solar minima 2008 and maxima 2013.

| Year | 2008 | | | | | | | | | 2013 | | | | | | | | |
|---|---|---|---|---|---|---|---|---|---|---|---|---|---|---|---|---|---|---|
| | Statistical Parameters | | | | | | | | | Statistical Parameters | | | | | | | | |
| | RMSE | | | Bias | | | R | | | RMSE | | | Bias | | | R | | |
| Months | Min | Max | Mean | Min | Max | Mean | Min | Max | Mean | Min | Max | Mean | Min | Max | Mean | Min | Max | Mean |
| Jan | 0.77 | 6.53 | 3.09 | -5.13 | 1.79 | -2.02 | -0.32 | 0.97 | 0.78 | 1.74 | 12.4 | 6.96 | -11.04 | 6.7 | -4.16 | -0.22 | 0.96 | 0.8 |
| Feb | 1.05 | 7.42 | 3 | -5.01 | 2.12 | -1.99 | -0.07 | 0.98 | 0.86 | 1.44 | 11.61 | 4.84 | -8.11 | 8.43 | -1.63 | -0.16 | 0.98 | 0.85 |
| Mar | 1.27 | 8.93 | 3.64 | -7.13 | 2 | -2.58 | 0.33 | 0.97 | 0.88 | 2.22 | 16.59 | 6.48 | -11.77 | 5.31 | -3.61 | 0.13 | 0.97 | 0.86 |
| Apr | 1.14 | 8.71 | 3.35 | -6.69 | 2.53 | -2.23 | 0 | 0.98 | 0.86 | 3.07 | 20.48 | 8.04 | -15.06 | 7.87 | -5.11 | -0.06 | 0.96 | 0.83 |
| May | 0.66 | 9.82 | 2.91 | -5.69 | 4.17 | -1.51 | -0.08 | 0.98 | 0.85 | 1.68 | 18.78 | 7.6 | -16.13 | 6.59 | -5.55 | -0.16 | 0.98 | 0.84 |
| Jun | 0.58 | 5.92 | 2.34 | -4.68 | 1.36 | -1.49 | -0.14 | 0.98 | 0.8 | 0.9 | 12.44 | 5.28 | -10.01 | 5.34 | -3.45 | -0.21 | 0.98 | 0.8 |
| Jul | 0.48 | 5.91 | 2.16 | -4.51 | 1.35 | -1.37 | -0.17 | 0.98 | 0.82 | 0.74 | 11.85 | 4.78 | -9.65 | 4.39 | -2.79 | -0.14 | 0.99 | 0.82 |
| Aug | 0.56 | 8.31 | 2.35 | -3.87 | 2.52 | -1.22 | 0.32 | 0.98 | 0.88 | 1.36 | 14.92 | 5.24 | -11.73 | 5.93 | -2.44 | 0.11 | 0.98 | 0.86 |
| Sep | 0.61 | 11.42 | 2.77 | -3.9 | 4.69 | -0.89 | 0.71 | 0.98 | 0.91 | 1.58 | 12.9 | 5.17 | -6.72 | 8.92 | 0.04 | -0.05 | 0.98 | 0.88 |
| Oct | 0.76 | 11.63 | 3.09 | -3.87 | 6.21 | -0.76 | 0.36 | 0.98 | 0.89 | 2.49 | 20.81 | 7.52 | -13.69 | 6.02 | -3.77 | -0.05 | 0.98 | 0.83 |
| Nov | 0.78 | 8.46 | 3.08 | -4.35 | 5.83 | -0.91 | -0.1 | 0.98 | 0.81 | 2.72 | 24.27 | 9.33 | -16.54 | 5.34 | -6.02 | 0 | 0.97 | 0.82 |
| Dec | 0.46 | 7.31 | 2.8 | -4.8 | 3.97 | -1.12 | -0.38 | 0.98 | 0.78 | 2.19 | 21.15 | 9.39 | -18.2 | 4.99 | -6.4 | -0.35 | 0.97 | 0.8 |

### 3.1.2 Comparison of IRI-2016 simulation and GPS-TEC observations at selected longitude sectors

Fig. 4 shows scatter plots of TECs from IRI-2016 versus IGS GPS at 4 selected longitudes. It has been already noted in Section 3.1.1 that performance of the IRI-2016 model degrades with high solar activity and summer season of each hemisphere. The IRI-2016 TEC at the four selected longitudes is low biased (1.7 to 1.9 TECU) against GPS-TEC during 2008 (Fig. 4, four

left panels). In contrast, this bias has increased to values ranging from 3.7 to 4.5 TECU (four right panels of Fig. 4) during 2013 consistent with analysis on the monthly time scale for the whole globe. The pattern observed in bias is similar for RMSE which increased from lowest value of 1.9 TECU in 2008 to highest value of 5.6 TECU in 2013. Similarly, the correlation dropped from highest 0.97 in 2008 to lowest 0.93 in 2013. Much of the discrepancies are attributed to weaker performance of the model during summer of each hemisphere at these longitudes. However data points from EIA crest regions along these

longitudes have also contributed to the high bias and RMSE as well as low correlations. The performance of the model in capturing daytime TEC is better than nighttime TEC (not shown). The presence of large scatter at lower and higher ends of the distribution is common during both solar activity years. At both ends, IRI-TEC is low biased (See Fig. 4). The level of weakness in IRI-2016 at these parts of TEC distribution is further assessed in Sections 3.2-3.3 using quantile-based categorical metrics.

### 3.1.3 Comparison of IRI-2016 simulation and GPS-TEC observations on a seasonal basis

In previous sections, the comparisons were based either on individual data within a given calendar month (Section 3.1.1) or the whole year (Section 3.1.2). However, as we have noted in these sections, there is indication that the model performance is a function of local time and seasons. Therefore, the calendar months are grouped into four seasons (namely March Equinox, June Solstice, September Equinox and December Solstice) and the RMSE, bias and correlation between IRI TEC and GPS

TEC are determined on a seasonal time scale. The RMSE, Bias and correlation of TECs from the model and GPS are depicted





**Figure 2.** Monthly Comparison (January (top-left) to December (bottom-right)) of TEC from IRI-2016 Model and IGS GPS during solar minima 2008.





**Figure 3.** Monthly Comparison (January (top-left) to December (bottom-right)) of TEC from IRI-2016 Model and IGS GPS during solar maxima 2013.





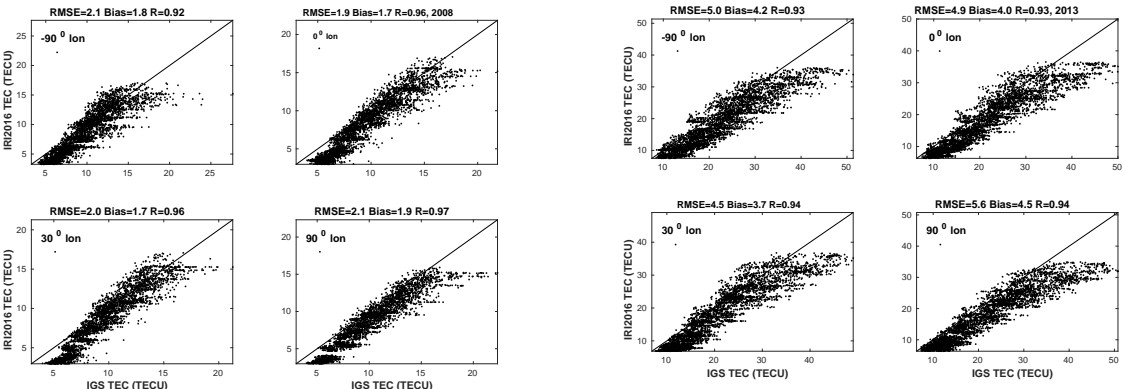

**Figure 4.** Scatter plots of TECs from IRI-2016 and GPS at selected longitudes for solar minimum 2008 (first two columns) and solar maximum 2013 (last two columns). RMSE, Bias and correlation (R) are indicated at the top of each panels.

in Fig. 5 for 2008 and in Fig. 6 for 2013 period respectively.

In 2008 (solar minimum period), the RMSE is generally higher during March and September equinoctial months than other seasons over tropics (Fig. 5). However, pronounced positive biases mainly along EIA crest regions are noted during September

Equinox and December Solstice whereas IRI-TEC is mainly low biased against GPS TEC over the rest of the globe. In contrast, correlations are generally high in all seasons over most of the globe with the exception in March Equinox, June Solstice and December Solstice over southern Atlantic, Pacific oceans and the polar regions which had low correlation between IRI-TEC and GPS-TEC.

In 2013 (solar maximum period), RMSE has generally increased over tropics from values in 2008 and over the whole of southern hemisphere (during September Equinox and December Solstice) and northern hemisphere (during March Equinox and June Solstice) (Fig. 6). Moreover, the high negative bias dominated southern hemisphere during December Solstice (upto -16 TECU) and northern hemisphere during June Solstice (upto -12 TECU). The phase of TEC variation is well captured by the IRI-2016 model with some differences between seasons as revealed from weaker correlation southward of $50^0$S during

December Solstice, southward of $30^0$S during September Equinox and northward of $30^0$N during June Solstice than during March Equinox in 2013. This confirms that relative to correlation between IRI-TEC and GPS-TEC in 2008, there is a decrease in correlation over southern mid-latitude ionosphere. Table 3 lists the global minimum, maximum and mean of RMSE, bias and correlation during the four seasons of 2008 and 2013. The lowest of the global minimum RMSE (0.59 TECU) is observed during June Solstice in 2008. In contrast, the lowest of global minimum RMSE of 1.24 TECU is also observed during June

solstice in 2013. The highest of global maximum RMSE is 11.5 TECU during September Equinox and 22.78 TECU during December Solstice in 2008 and 2013 respectively. The global mean RMSE is within 2.46 to 3.53 TECU in 2008. In contrast,





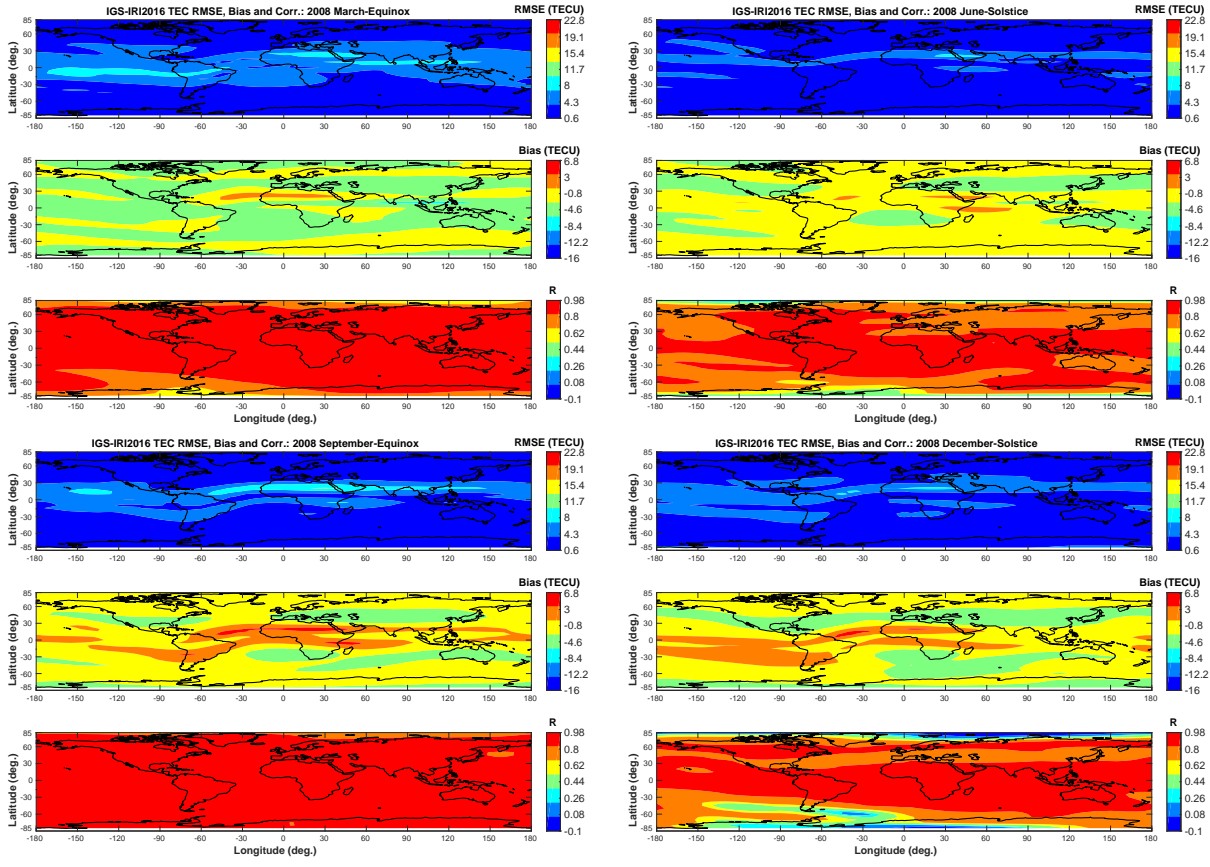

**Figure 5.** Seasonal variations [March-Equinox (top left panels), June-Solstice (top right panels), September-Equinox (bottom left panels), December-Solstice (bottom right panels) ] of TEC from IRI-2016 model against GPS TEC during solar minima year 2008.

**Table 3.** Statistical parameters for seasonal variations of solar minima 2008 and maxima 2013.

| Year | 2008 | | | | | | | | | | | | 2013 | | | | | | | | | | | |
|---|---|---|---|---|---|---|---|---|---|---|---|---|---|---|---|---|---|---|---|---|---|---|---|---|
| Seasons | March Equinox | | | June Solstice | | | September Equinox | | | December Solstice | | | March Equinox | | | June Solstice | | | September Equinox | | | December Solstice | | |
| Stat. Par. | RMSE | Bias | R | RMSE | Bias | R | RMSE | Bias | R | RMSE | Bias | R | RMSE | Bias | R | RMSE | Bias | R | RMSE | Bias | R | RMSE | Bias | R |
| Min | 1.26 | -7.01 | 0.58 | 0.59 | -4.68 | 0.22 | 0.72 | -3.86 | 0.75 | 0.68 | -4.38 | -0.1 | 3.34 | -13.3 | 0.45 | 1.24 | -11.17 | -0.12 | 2.19 | -9.49 | 0.09 | 2.61 | -16.36 | 0.04 |
| Max | 8.56 | 2.01 | 0.97 | 7.7 | 2.28 | 0.98 | 11.5 | 5.37 | 0.98 | 7.89 | 4.83 | 0.97 | 18.03 | 6.7 | 0.96 | 14.34 | 5.31 | 0.97 | 16.89 | 7.06 | 0.97 | 22.78 | 5.02 | 0.96 |
| Mean | 3.53 | -2.43 | 0.88 | 2.46 | -1.4 | 0.85 | 2.94 | -0.83 | 0.92 | 2.95 | -1 | 0.8 | 7.4 | -4.41 | 0.85 | 5.9 | -3.55 | 0.83 | 6.44 | -1.74 | 0.85 | 9.36 | -6.16 | 0.81 |

in 2013, it varies from 5.9 during June Solstice to 9.36 TECU during December Solstice. The global average bias ranges from -0.83 to -2.43 TECU in 2008 whereas it varies from -1.74 TECU during September Equinox to -6.16 TECU during December Solstice (See Table 3). The correlation is within a range of -0.1 to 0.98 with a spatial mean value of 0.8 to 9.2 in 2008. The lowest range of correlation (0.75 to 0.98) is during September Equinox in 2008. Similarly the lowest range of correlation
5 (0.45 to 0.96) in 2013 is during March Equinox as compared to other seasons. The global mean correlations are 0.81 during December Solstice and 0.85 during March and September Equinoctial months in 2013 (see Table 3).





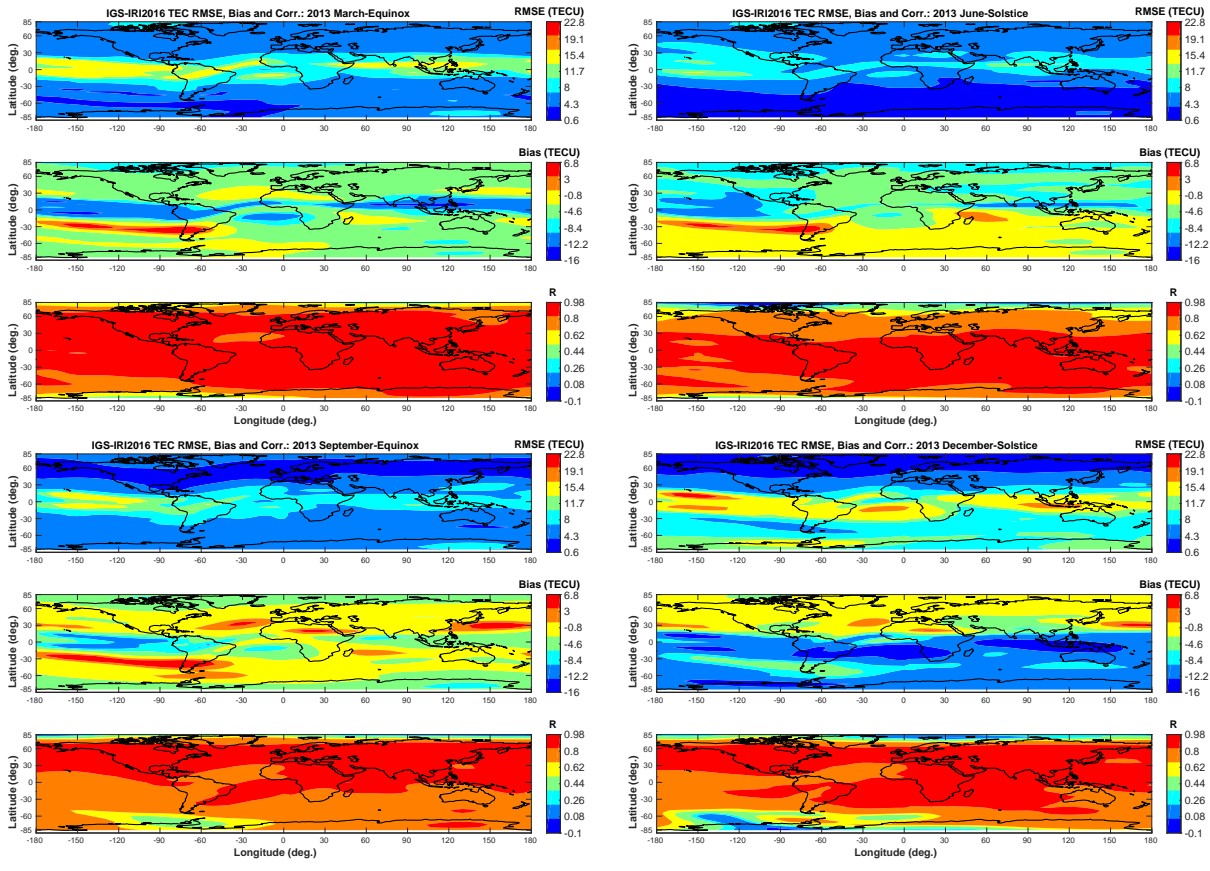

**Figure 6.** Seasonal variations [March-Equinox (top left panels), June-Solstice (top right panels), September-Equinox (bottom left panels), December-Solstice (bottom right panels) ] of TEC from IRI-2016 model against GPS TEC during solar maxima year 2013.

## 3.2 Categorical Statistics: QPOD, QFAR, QCSI, QCM

### 3.2.1 Categorical comparison of IRI-2016 simulation and GPS-TEC observations

As noted in Section 3.1.2 with the scatter plots at lower and upper ends of TEC distribution, most of the deviations from observations arise at these ends. Therefore, there has been efforts to understand these discrepancies. For instance, Venkata Ratnam
et al. (2017) have included relative TEC deviation index, monthly variations in the grand mean of ionospheric TEC, TEC intensity, the upper and lower quartiles in their comparison of GPS-TEC with IRI-2007 and IRI-2012 predicted TECs. Although the inclusion of lower and upper quartiles is a step in the right direction to understand the discrepancy in these parts of the distribution, much of the observed differences lie in the extreme ends within the quartiles. Therefore, application of quantile categorical statistics is necessary for more insight into the problem as indicated in Section 3.2. Since much of the data used to constrain the IRI-model represent mainly the mean values, the model is likely to under perform at the extreme ends of TEC





distribution. Therefore, QPOD, QFAR, QCSI and QCM are employed in this section to assess the performance of IRI-2016 against GPS-TEC observations.

Fig. 7 shows QPOD, QFAR, QCM, and QCSI for TEC values exceeding 10, 25, 75 and 90 percentiles for 2008. The notable
feature in Fig. 7 is the decrease in QPOD as the percentile increases from (Fig. 7a) to (Fig. 7m) over mid-and polar latitudes suggesting that the model skill decreases as the high extremes are dominant parts of TEC values in the evaluation of the metrics. However, these trends in quantile metrics are reversed with increase in threshold percentiles from 10% to 25% implying model performs also weakly at the low extreme. Similar changes in values of metrics that measure model skill is observed during 2013 (see Fig. 8). Consistent with this, QCM (Fig. 7c, 7g, 7k and 7o) exhibits increasing trend with percentile increase
as expected. In contrast, QCSI increases global as the percentile changes from 10% (Fig. 7d) to 25% (Fig. 7h) globally. Moreover, QCSI begins to decrease with a change from 25% (Fig. 7h) to 90% (Fig. 7p) globally. The difference in pattern between QPOD and QCSI is attributed to the fact that the false alarm rate has increased with increase in percentile threshold from 25% (Fig. 7f) to 90%(Fig. 7n). In particular, the increase in false alarm rate of detection of observations by the simulation over the EIA crest region is quite evident with a shift from $10^{th}$ percentile to $90^{th}$ percentile in 2008. This false model skill has been
removed in QCSI as opposed to QPOD which shows high model skill. Similar patterns of metrics that measure model skill is observed during 2013 (Fig. 8). However, the rate of decrease from 25 to 90 percentile is significantly higher than those in 2008 in particular over southern hemisphere. For example QCSI drops from 1 (perfect skill) for 10 percentile to about 0.4 for 90 percentile over tropics covering EIA crest region. This is consistent with results of Section 3.1 that shows weakness in the IRI-2016 model during enhanced solar activity. Therefore we noticed here generally the IRI-2016 model has better agreement
with GPS during solar minima 2008 than solar maxima 2013 at the extreme margins of TEC distribution.

There are also some major notable characteristics that can be highlighted from individual categorical metrics. QPOD varies from 0.4 to 0.8 over tropics at $10^{th}$ percentiles with maximum mainly along the northern EIA crest regions during 2008 solar minimum year (Fig. 7a). At $25^{th}$ percentile (Fig. 7e), the change in QPOD from $10^{th}$ percentile is notable only over the
northern EIA crest region (increase from 0.8 to 1). However, at the $75^{th}$ percentile (Fig. 7i), significant drop in QPOD ( below 0.2) over northern mid and high latitudes, magnetic equator along Asian sector and southern polar region is observed. At the same time, the model skill in capturing observed TEC over southern mid-latitude and tropical ionosphere is enhanced with QPOD values in the range of 0.7 to 1. This skill of the model at $90^{th}$ percentile has persisted over EIA crest regions with drop in QPOD elsewhere (Fig. 7m). The overall patterns of QCM ( i.e., low over tropics except magnetic equator versus high
over high latitudes) and QCSI( i.e., high over tropics except magnetic equator versus low over high latitudes) at the $75^{th}$ and $90^{th}$ percentile threshold levels are consistent with QPOD and QFAR changes at this threshold level. The difference in spatial patterns between these metrics is hardly apparent except at transition zones between mid- and high latitudes during 2008 solar minimum year.



**Table 4.** Categorical comparison of statistical parameters for solar minima 2008 and maxima 2013.

| Year | 2008 | | | | | | | | | | | | 2013 | | | | | | | | | | | |
|---|---|---|---|---|---|---|---|---|---|---|---|---|---|---|---|---|---|---|---|---|---|---|---|---|
| Parameter | QPOD | | | QFAR | | | QCM | | | QCSI | | | QPOD | | | QFAR | | | QCM | | | QCSI | | |
| Percentile | Min | Max | Mean | Min | Max | Mean | Min | Max | Mean | Min | Max | Mean | Min | Max | Mean | Min | Max | Mean | Min | Max | Mean | Min | Max | Mean |
| 10% | 0.47 | 0.99 | 0.71 | 0 | 0.09 | 0.003 | 0.007 | 0.53 | 0.29 | 0.47 | 0.93 | 0.71 | 0.68 | 1 | 0.89 | 0 | 0.10 | 0.02 | 0 | 0.32 | 0.11 | 0.68 | 0.97 | 0.87 |
| 25% | 0.50 | 0.97 | 0.75 | 0 | 0.08 | 0.01 | 0.03 | 0.50 | 0.25 | 0.50 | 0.94 | 0.75 | 0.54 | 1 | 0.87 | 0 | 0.15 | 0.03 | 0 | 0.46 | 0.13 | 0.54 | 0.95 | 0.84 |
| 75% | 0 | 1 | 0.49 | 0 | 0.80 | 0.14 | 0.001 | 1 | 0.51 | 0 | 0.84 | 0.43 | 0 | 0.98 | 0.35 | 0 | 1 | 0.22 | 0.02 | 1 | 0.65 | 0 | 0.74 | 0.30 |
| 90% | 0 | 1 | 0.29 | 0 | 1 | 0.36 | 0.002 | 1 | 0.71 | 0 | 0.63 | 0.19 | 0 | 0.89 | 0.13 | 0 | 1 | 0.43 | 0.11 | 1 | 0.87 | 0 | 0.53 | 0.10 |

In contrast to 2008, the model performed very well in capturing observed TEC above $10^{th}$ percentile (Fig. 8a) in 2013 over most of the ionosphere as evidenced by high QPOD above 0.8. However, the model skill continued to be poor over southern parts of North America and Europe ionosphere as well as over areas from Australia to South Africa. The model skill improved with increase in percentile thresholds (10% (Fig. 8a) to 25% (Fig. 8e)) globally in 2013. At $75^{th}$ (Fig. 8i) and $90^{th}$ (Fig.

8m) percentiles, the skill of the model deteriorated over tropics with the exception of a few longitude sectors along northern EIA crest region from its performance at $25^{th}$ percentile. The model is also deteriorated further over magnetic equator, mid- and high latitudes (Fig. 8). Other metrics (i.e., QFAR, QCM and QCSI) portray similar features. Table 4 summarizes global minimum, maximum, and mean of QPOD, QFAR, QCM and QCSI at all percentile levels for the two periods under study. The lowest skill as demonstrated by QPOD and QCSI of zero is noted at the $75^{th}$ and $90^{th}$ percentiles in 2008 and 2013.

The highest QFAR and QCM of one are attained at $90^{th}$ percentile during 2008 and at $75^{th}$ percentile during 2013. The mean QPOD varies from 0.29 at $90^{th}$ to 0.71 at $10^{th}$ percentiles during 2008. In contrast, QPOD varies from 0.13 at $90^{th}$ to 0.89 at $10^{th}$ percentiles. These features are also exhibited by QCSI as it varies from global mean of 0.19 and 0.10 at $90^{th}$ to 0.71 and 0.87 at $10^{th}$ percentiles during 2008 and 2013 respectively. The global mean QFAR increases from 0 at $10^{th}$ to 0.36 and 0.43 at $90^{th}$ percentiles during 2008 and 2013 respectively. Similarly, global mean QCM increases from 0.29 and 0.11 at $10^{th}$ to

0.71 and 0.87 at the $90^{th}$ percentiles in the same order (see Table 4).

### 3.3 Categorical comparison of IRI-2016 simulation and GPS-TEC observations on a seasonal basis

The categorical metrics for the four seasons (March Equinox, June Solstice, September Equinox and December Solstice) are given in Figs. 9-11 for 2008 and 2013.

Fig. 9 depicts QPOD for March Equinox (Fig. 9a-d), June-Solstice (Fig. 9e-h), September-Equinox (Fig. 9i-l) and December-Solstice (Fig. 9m-p) during 2008 (first group of columns) and 2013 (last group of columns) at $10^{th}$, $25^{th}$, $75^{th}$ and $90^{th}$ percentiles. During 2008 (March Equinox season), IRI-2016 model has identified over 80% of GPS-TEC exceeding $10^{th}$ percentile over broader areas along geomagnetic equator and at isolated places south of Australia (Fig. 9a) correctly. This figure remained the same at $25^{th}$ percentile (Fig. 9b). However, QPOD increased to values exceeding 60% over the rest of tropics

and mid-latitudes. IRI-2016 captures more than 85% of observed TEC exceeding $75^{th}$ and $90^{th}$ percentiles over the EIA crest region and exhibits a steady drop (to less than 20%) in skill over the rest of the globe. Specifically, the change in QPOD along geomagnetic equator from values exceeding 80% at the $10^{th}$ percentile to values less than 20% at the $75^{th}$ percentile is signif-





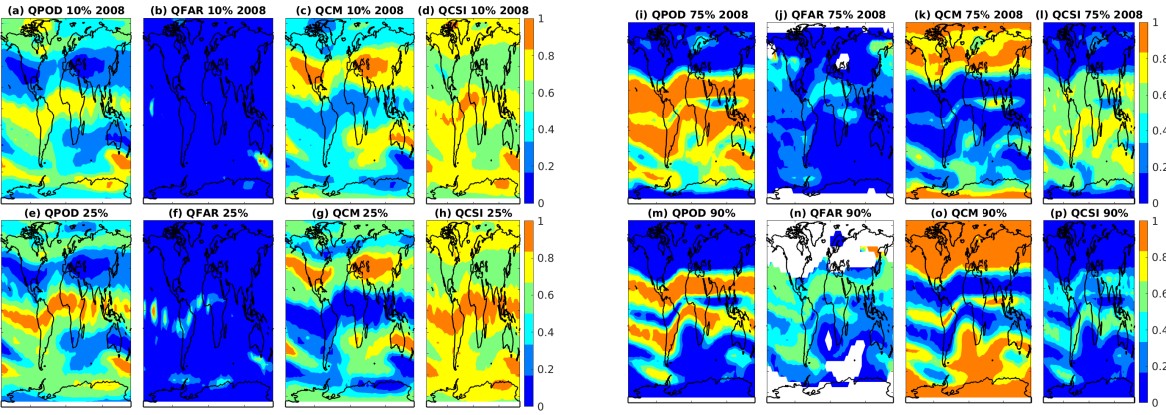

**Figure 7.** Categorical variation of TEC [10% ( (a)-(d)), 25% ( (e)-(h) ), 75% ((i)-(l)), 90% ((m)-(p)) ] from IRI-2016 model against GPS TEC during solar minima year 2008.

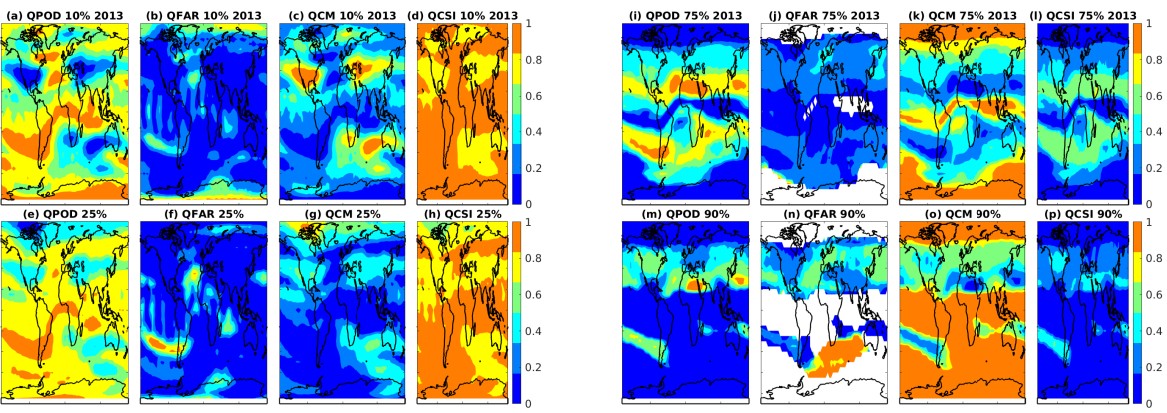

**Figure 8.** Categorical variation of TEC [10% ((a)-(d)), 25% ((e)-(h)), 75% ((i)-(l)), 90% ((m)-(p))] from IRI-2016 model against GPS TEC during solar maxima year 2013.



icant. The detection skill of IRI model continued to decrease as the threshold increased from $75^{th}$ to $90^{th}$ percentile(Fig. 9c-d, left panels). Since much of the night time and day time TECs constitute low and high extremes in the climatology of TEC, it is consistent with previous understanding to observe decline in model skill over magnetic equator as percentile threshold increases during 2008. Conversely, the performance of the model during June solstice in identifying the lower ends of TEC

distribution (values higher than $10^{th}$ percentile) has greatly declined with a score of 60-80% over geomagnetic equator (Fig. 9e, left). However, at $25^{th}$ percentile (Fig. 9f, left), QPOD increased over magnetic equator and northern Atlantic region to a value of 80-100%. The QPOD at $75^{th}$ and $90^{th}$ percentiles (Fig. 9g-h, left) decreased significantly over magnetic equator, northern mid- and high latitudes. In contrast, there is improvement in model skill over EIA crest region at these thresholds. The skill of the model remained poor at all thresholds over southern polar regions during June solstice (Fig. 9e-h, left). The performance

of the model during September Equinox is shown in Fig. 9i-l. At the $10^{th}$ percentile (Fig. 9i, left), QPOD over geomagnetic equator and high latitudes exceeds 80% whereas much of northern mid and high latitudes, as well as areas between tip of South Africa and Australia exhibit poor model skill having QPOD of less than 20%. At $25^{th}$ percentile (Fig. 9j, left), there is a significant increase in QPOD over much of the globe. However, at $75^{th}$ and $90^{th}$ percentiles (Fig. 9k-l), a decrease in QPOD over northern mid latitude, polar regions and geomagnetic equator was observed. In contrast there is an increase in the model

skill over EIA regions. During the December solstice of 2008 at the $10^{th}$ and $25^{th}$ percentiles (Fig. 9m-n,left), the QPOD is within the range of 60-100% over geomagnetic equator, southern mid and high latitudes whereas the QPOD over most of northern mid and high latitudes is within 20-40%. On the other hand, at $75^{th}$ and $90^{th}$ percentiles (Fig. 9o-p, left), QPOD improves to a value exceeding 80% over EIA crest regions while it exhibits decrease over both northern and southern mid-and high latitudes. The changes over the southern mid and high latitudes during this season at the $75^{th}$ and $90^{th}$ percentiles appear

to be a mirror reflection of June solstice in the northern hemisphere. This similarity in model detection skill during the two solstices is also apparent at the lower ends of TEC distribution. Unlike June solstice, the higher skill score covered most of the southern hemisphere(see Fig. 9m, left).

In 2013 during solar maximum year, the QPOD characteristics are similar to that of 2008 for all the seasons but with notable

improvement at the lower ends for the two equinoctial seasons (see Fig. 9a-b and Fig. 9i-j, right). In contrast to 2008, the model detection skill at the $75^{th}$ and $90^{th}$ percentiles has weakened over the EIA crest regions. Instead, improved performance of IRI-2016 model can be seen over most of southern and northern hemispheres during December and June solstices respectively (Fig. 9g-h and Fig. 9o-p, right). Unlike the solstices, during March and September equinoctial months, the performance at the $75^{th}$ percentile is good across broader areas along EIA crest regions and hemispherically symmetric (Fig. 9g,h, right). At the

$90^{th}$ percentile, the model performance is very bad over most parts of the globe during March Equinox and reasonably good over northern mid-latitude during September Equinox (Fig. 9d,l, right). This suggests that the observed TEC distribution has slightly shifted towards higher values relative to 2008 as a whole which is consistent with the high solar activity. This conclusion follows from the fact that any improvement in model performance arises from the nature of the observed TEC distribution rather than the model itself since the model configuration remained the same. This changes in skill is also apparent within the

same year from one season to the other as noted in previous paragraph. Table 5 summarizes the changes in QPOD with season




**Table 5.** Statistical parameters of QPOD for all seasonal variations of solar minima 2008 maxima 2013.

| Year | 2008 | | | | | | | | | | | | 2013 | | | | | | | | | | | |
|---|---|---|---|---|---|---|---|---|---|---|---|---|---|---|---|---|---|---|---|---|---|---|---|---|
| Parameter | QPOD | | | | | | | | | | | | QPOD | | | | | | | | | | | |
| Seasons | March Equinox | | | June Solstice | | | September Equinox | | | December Solstice | | | March Equinox | | | June Solstice | | | September Equinox | | | December Solstice | | |
| Percentile | Min | Max | Mean | Min | Max | Mean | Min | Max | Mean | Min | Max | Mean | Min | Max | Mean | Min | Max | Mean | Min | Max | Mean | Min | Max | Mean |
| 10% | 0.3 | 0.93 | 0.64 | 0 | 1 | 0.58 | 0.44 | 0.99 | 0.71 | 0 | 1 | 0.64 | 0 | 1 | 0.79 | 0 | 1 | 0.72 | 0 | 1 | 0.87 | 0 | 1 | 0.72 |
| 25% | 0 | 0.97 | 0.64 | 0 | 0.99 | 0.56 | 0.29 | 1 | 0.73 | 0 | 1 | 0.64 | 0 | 1 | 0.75 | 0 | 1 | 0.65 | 0 | 1 | 0.85 | 0 | 1 | 0.66 |
| 75% | 0 | 1 | 0.32 | 0 | 1 | 0.42 | 0 | 1 | 0.55 | 0 | 1 | 0.52 | 0 | 1 | 0.32 | 0 | 1 | 0.44 | 0 | 1 | 0.53 | 0 | 1 | 0.32 |
| 90% | 0 | 1 | 0.14 | 0 | 1 | 0.36 | 0 | 1 | 0.39 | 0 | 1 | 0.39 | 0 | 1 | 0.07 | 0 | 1 | 0.32 | 0 | 1 | 0.27 | 0 | 1 | 0.21 |

and solar activity. The spatial minimum, maximum and mean of QPOD at all percentile levels are indicated for the two solar activity periods. The minimum and maximum of spatial mean QPOD at $10^{th}$ percentile have occurred during June solstice and September equinox of 2008 respectively. However, the minimum and maximum of spatial mean QPOD have been observed during December solstice and September equinox of 2013 respectively. The lowest spatial mean QPOD at the $90^{th}$ percentile is observed during March Equinox of 2008 and 2013 whereas the highest QPOD is during September equinox and September solstices in 2008 and during June solstice in 2013.

Fig. 10 shows QCM at $10^{th}$, $25^{th}$, $75^{th}$ and $90^{th}$ percentiles for 2008 (left panels) and 2013 (right panels) for the four seasons as in Fig. 9. As QCM quantifies TEC observed by GPS but missed by IRI-2016 model, a perfect score is given by QCM value of zero whereas zero skill in model is expressed with QCM of one. Therefore, the spatial patterns observed for QPOD are expected to match that of categorical miss as shown in Fig. 10. QCM is maximum over high and polar latitudes during June solstice, March and September equinoctial seasons whereas tropics is characterized by low values of QCM in general which are consistent with QPOD features. Table 6 provides overview of the statistics of QCM for the four seasons and the two solar activity periods.

Fig. 11 shows QCSI at the four percentile levels during 2008 (left panels) and 2013 (right panels). The model performance as assessed by QCSI remains the same as QPOD at $10^{th}$ and $25^{th}$ percentiles (compare Figs. 9-10 with Fig. 11). However, IRI-2016 performance, as revealed from QCSI values, differ from that suggested by QPOD at $75^{th}$ percentile during December solstice and at $90^{th}$ percentiles during all seasons (Fig. 11o, d, h, l, p, left). This is due to the fact that QCSI combines QPOD and QFAR features to describe the skill of the model in more robust manner. However, QCSI shows slightly higher model skill during 2013 at the lower ends as confirmed by high value of QCSI ranging from 80% to 100% over most of the globe. Moreover, the skill at the high extreme tail of the TEC distribution during 2013 is relatively weaker than its performance in 2008 (Fig. 11 d, h, l, p, right). Table 7 summarizes the globally averaged mean of QCSI and its extremes during 2008 and 2013 for the four seasons. Clearly, the performance of IRI-2016 at the low extreme tail is better in 2013 than in 2008. This has been reversed at the high extreme portion ( $90^{th}$ percentile) of the TEC distribution with lower QCSI in 2013 than in 2008.





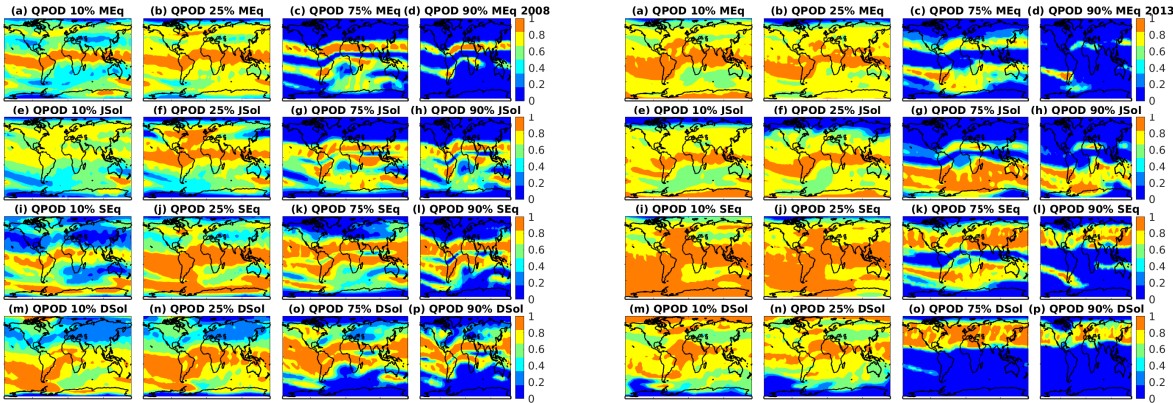

**Figure 9.** QPOD for all seasonal variations of TEC Percentile description of extreme distributions (10%, 25%, 75% and 90% ) from IRI-2016 model against GPS-TEC during solar minima 2008 (left panels) and solar maxima 2013 (right panels).

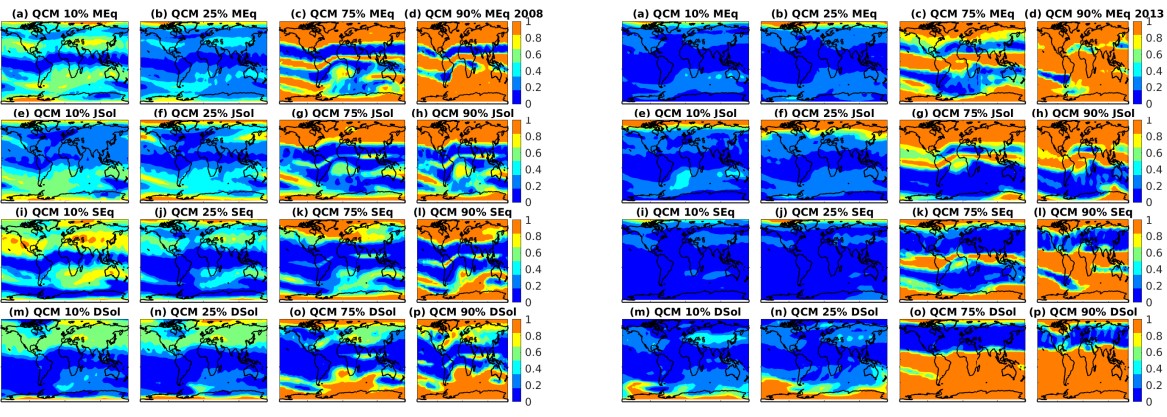

**Figure 10.** QCM for all seasonal variations of TEC Percentile description of extreme distributions (10%, 25%, 75% and 90% ) from IRI-2016 model against GPS-TEC during solar minima 2008 (left panels) and solar maxima 2013 (right panels).

**Table 6.** Statistical parameters of QCM for all seasonal variations of solar minima 2008 and maxima 2013.

| Year | 2008 | | | | | | | | | | | | 2013 | | | | | | | | | | | |
|---|---|---|---|---|---|---|---|---|---|---|---|---|---|---|---|---|---|---|---|---|---|---|---|---|
| Parameter | QCM | | | | | | | | | | | | QCM | | | | | | | | | | | |
| Seasons | March Equinox | | | June Solstice | | | September Equinox | | | December Solstice | | | March Equinox | | | June Solstice | | | September Equinox | | | December Solstice | | |
| Percentile | Min | Max | Mean | Min | Max | Mean | Min | Max | Mean | Min | Max | Mean | Min | Max | Mean | Min | Max | Mean | Min | Max | Mean | Min | Max | Mean |
| 10% | 0.07 | 0.7 | 0.36 | 0 | 1 | 0.42 | 0.01 | 0.56 | 0.29 | 0 | 1 | 0.36 | 0 | 1 | 0.21 | 0 | 1 | 0.28 | 0 | 1 | 0.13 | 0 | 1 | 0.28 |
| 25% | 0.03 | 1 | 0.36 | 0.01 | 1 | 0.44 | 0 | 0.71 | 0.27 | 0 | 1 | 0.36 | 0 | 1 | 0.25 | 0 | 1 | 0.35 | 0 | 1 | 0.15 | 0 | 1 | 0.34 |
| 75% | 0 | 1 | 0.68 | 0 | 1 | 0.58 | 0 | 1 | 0.45 | 0 | 1 | 0.48 | 0 | 1 | 0.68 | 0 | 1 | 0.56 | 0 | 1 | 0.47 | 0 | 1 | 0.68 |
| 90% | 0 | 1 | 0.86 | 0 | 1 | 0.64 | 0 | 1 | 0.61 | 0 | 1 | 0.61 | 0 | 1 | 0.93 | 0 | 1 | 0.68 | 0 | 1 | 0.73 | 0 | 1 | 0.79 |



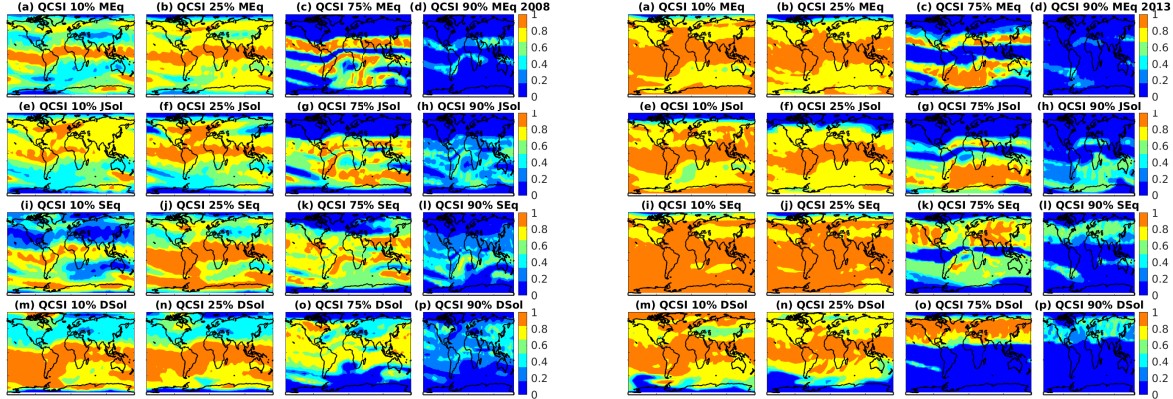

**Figure 11.** QCSI for all seasonal variations of TEC Percentile description of extreme distributions (10%, 25%, 75% and 90% ) from IRI-2016 model against GPS-TEC during solar minima 2008 (left panels) and solar maxima 2013 (right panels).

**Table 7.** Statistical parameters of QCSI for all seasonal variations of solar minima 2008 and maxima 2013.

| Year | 2008 | | | | | | | | | | | | 2013 | | | | | | | | | | | |
|---|---|---|---|---|---|---|---|---|---|---|---|---|---|---|---|---|---|---|---|---|---|---|---|---|
| Parameter | QCSI | | | | | | | | | | | | QCSI | | | | | | | | | | | |
| Seasons | March Equinox | | | June Solstice | | | September Equinox | | | December Solstice | | | March Equinox | | | June Solstice | | | September Equinox | | | December Solstice | | |
| Percentile | Min | Max | Mean | Min | Max | Mean | Min | Max | Mean | Min | Max | Mean | Min | Max | Mean | Min | Max | Mean | Min | Max | Mean | Min | Max | Mean |
| 10% | 0.3 | 0.93 | 0.63 | 0 | 0.95 | 0.58 | 0.44 | 0.96 | 0.71 | 0 | 0.97 | 0.63 | 0 | 0.98 | 0.77 | 0 | 0.97 | 0.7 | 0 | 0.98 | 0.84 | 0 | 0.97 | 0.7 |
| 25% | 0 | 0.95 | 0.63 | 0 | 0.93 | 0.55 | 0.29 | 0.97 | 0.72 | 0 | 0.97 | 0.62 | 0 | 0.99 | 0.73 | 0 | 0.97 | 0.63 | 0 | 0.99 | 0.8 | 0 | 0.97 | 0.62 |
| 75% | 0 | 0.89 | 0.28 | 0 | 0.87 | 0.36 | 0 | 0.83 | 0.44 | 0 | 0.91 | 0.42 | 0 | 0.77 | 0.26 | 0 | 0.86 | 0.35 | 0 | 0.86 | 0.4 | 0 | 0.89 | 0.25 |
| 90% | 0 | 0.66 | 0.08 | 0 | 0.68 | 0.23 | 0 | 0.81 | 0.22 | 0 | 0.67 | 0.22 | 0 | 0.5 | 0.04 | 0 | 0.74 | 0.21 | 0 | 0.81 | 0.17 | 0 | 0.7 | 0.13 |

## 4 Conclusions

In this paper, the performance of IRI-2016 model in simulating GPS-TEC is assessed employing RMSE, bias, correlation and categorical metrics such as Quantile Probability of Detection (QPOD), Quantile False Alarm Ratio (QFAR), Quantile Categorical Miss (QCM), and Quantile Critical Success Index(QCSI) during two distinct solar activity periods. The IRI-2016 model simulations are based on configuration that uses latest developments.

The monthly RMSE varies over a narrow range in 2008 as compared to 2013 with the upper part of the range over EIA crest regions. The summer hemispheres experience higher RMSE than winter hemisphere as demonstrated by high (low) RMSE from May to August over the northern (southern) hemisphere and vice-verse from October to January during solar maximum period of 2013. The bias in IRI-model prediction is negative over most parts of the globe with exception over EIA crest regions which are characterized by positive bias due to model overestimation. There is longitudinal variation in TEC bias leading to peak bias over Africa and American longitude sectors. These results are consistent with previous studies that show IRI model underestimates observed TEC at most places except the EIA crest regions. Some of these discrepancies are attributed to difference in altitudes between IRI model and GPS observations which results in plasmaspheric contribution in the GPS observation



and lack of plasmaspheric TEC in the IRI model according to several past studies. It is also evident that the IRI-2016 model captures the phase of TEC variation with great accuracy as revealed by high correlations over most of the globe.

The scatter plots at selected longitude sectors indicate that IRI-2016 model is low biased at both low and high tails of the

TEC distribution suggesting that IRI-2016 is capable of satisfactorily simulating the mean TEC globally. The extent of the IRI model weakness and strength at the extreme portions of observed TEC are assessed using categorical statistical metrics such as QPOD, QCSI, QFAR and QCM using $10^{th}$ and $25^{th}$ percentiles as lower margin and $75^{th}$ and $90^{th}$ percentiles as upper margins of the TEC distribution for the two distinct solar activity periods. The performance for the whole annual time series and seasonal time series were evaluated using these thresholds. The model has generally reasonable skill at the low ends of TEC

distribution over most of the globe. This skill weakens at high ends of the TEC distribution over much of the globe except EIA crest regions during both solar activity years. There is also hemispheric symmetry during June and December solstices with poorer performance over the summer hemisphere at the high extremes of observed TEC. This feature is consistent with high RMSE and low bias in model during summer as compared to winter time. Similarly, the robust skill at low ends of observed TEC distribution can be attributed to the fact that low TECs that constitute the low portion of TEC distribution are mainly

observed during night time while those at the high ends of the distribution occur during daytime.

In summary, the IRI-2016 model as a climatological empirical model have simulated significant portion of observed TEC with better accuracy during both solar activity periods and different seasons. The model performance at the extreme ends of the distribution is also remarkably good. In particular, the IRI-model skill in detecting observed TEC over EIA crest regions at

the extreme ends is robust despite high RMSE for the whole TEC distribution. Therefore, this encouraging IRI-2016 model performance at the extreme parts of observed TEC distribution suggests the importance of further work to improve the model so that it can be used for real time operational forecasting.

*Competing interests.* There is no conflict of interests.

*Acknowledgements.* We are highly grateful to NASA for free access to IRI-2016 model and GPS data. Moreover the first author extends his

gratitude to Aksum University, Addis Ababa University and Botswana International University of Science and Technology (BIUST) for their financial support during first author's PhD studies and research visit to BIUST.





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
