# Peer review of "Comparison of quiet time ionospheric total electron content from IRI-2016 model, gridded and station level GPS observations"

_Annales Geophysicae, 2019_

## Referee Comment (RC1) · Anonymous Referee #1 · 28 May 2019

The paper evaluated IRI-2016 model using GPS-TEC observations during the solar minima 2008 and maxima 2013. I understand, the authors have spent a lot of time and effort on model evaluation and paper writing. However, this paper is not organized in a logic way. The authors listed many figures in their paper but I find it difficult to understand which aspect of IRI2016 (for example: the performance in the EIA/the performance in hemispheric symmetry) they wanted to validate even though I have read their figures and the corresponding statements. I suggest the authors to rewrite it and emphasize the aspect you want to evaluate. Besides, I have some comments that need addressed in your revised version: Major 1. You used the TEC data extracted at a grid resolution of 5 latitude by 5 longitude from IGS. What is your time resolution? What

are the error distributions (spatial-temporal) of IGS TEC? How about the system error? I don't think the conclusions are convinced if the errors of the GPS-TEC are not clear. 2. The ability to simulate the diurnal variation of TEC is very important for a model. I suggest the authors to display some result about this performance of IRI2016. 3. Page 10, Section 3.1.2. You display some figures at 4 selected longitudes. But you didn't say anything about the difference between different longitudes. This may be an interesting work to do. 4. Page 8, Line7-11: I think it is not appropriate to conclude that the performance during high solar activity is poor just according to the large RMSE. As we know, the TEC is larger under high solar activity than under low solar activity. When the RMSE is analysed, the background value should not be ignored. The same situation should be considered when the RMSEs of different seasons and longitudes are analysed. 5. The authors should include a brief summary of the comparison between their conclusions and that of other recent publications in this field such as "Liu, Z. et al. (2019)" and "Acharya, R., & Majumdar, S. (2019)". Additionally, a literature review of the recent publications. Liu, Z., Fang, H., Weng, L., Wang, S., Niu, J., & Meng, X. (2019). A comparison of ionosonde measured foF2 and IRI-2016 predictions over China. Advances in Space Research, 63(6), 1926-1936. Acharya, R., & Majumdar, S. (2019). Comparison of observed ionospheric vertical TEC over the sea in Indian region with IRI-2016 model. Advances in Space Research, 63(6), 1892-1904. Minor 1. You used "S" and "O" to stand for simulated and observed data in Section 2.2.1. In Section 2.2.2, you used "SIM" and "OBS" for them, please confirm this and make them in accordance. 2. Page 7, Line 16: QPOD, not "POD". 3. Figure 4, the total number of points should be indicated. I usually do not like this kind of scatter plots that does not allow to appreciate the distribution of data. I prefer figures where the data density is more evident (e.g. using a color scale and binning the data in ranges).

---

## Referee Comment (RC2) · Anonymous Referee #2 · 15 Jul 2019

REVIEW OF THE ARTICLE Comparison of quite time ionospheric total electron content from IRI-2016 model and GPS observations Mulugeta Melaku and Gizaw Mengistu Tsidu

The article considers the real problem, namely the validity of the IRI-2016 ionosphere model, which has become the most popular model up to date. The subject is suitable for the Annales Geophysicae. However, I recommend the major revision and resubmission in order to improve the results. The following shortcomings were found:

1. In the abstract and in the first chapter the authors stated the requirement of the ionosphere for the correct forecast of the radio wave propagation. However, the

altitude range covered by IRI is suitable only for HF-UHF forecast. Moreover, TEC allows to predict only integral attenuation, as far as no profile properties can be derived from TEC. 2. The motivation for the article is not clearly stated. IRI has been verified intensively for years, using ionozondes, satellites and GPS receivers as well. What is the novelty of the work? What is the hypothesis to be checked using GPS observations and IRI calculation? Please state it clearly! 3. When the authors deal with 5x5 gridded TEC values they do not work with evidence. Instead they work with the results of an IGS computer model (some kind of Kalman filter and gridding technique). Thus, the title becomes wrong – you compare one model with another model. If they want to validate IRI model then exactly 422 sites must be used, with further gridding and mapping if necessary. 4. It is not clear, whether the authors used IRITEC subroutine, or they calculated vertical electron profiles and integrated them manually. 5. The monthly basis can suffer from biases. It is obvious to use 27 days periods corresponding to Bartels rotation cycles. 6. Short remark about (4). The correlation coefficient (4) makes sense only for stationary processes and for the processes that have normal distribution. No tests are presented that prove the aforementioned requirements. If they are violated then the results have no sense. 7. Figures 2 and 3 in Mercator projection are awful and unreadable. I see the authors want to prove that they have calculated everything declared. But at that scale it is impossible to make difference between Canada and the US. It is much better to choose a couple of the most interesting frames and print them at large scale. For high latitudes the orthogonal polar projection must be chosen. 8. The style of the presentation in the article can be accepted only if the authors used real F10.7 and Kp indices (or IG index) from the database. But even in this case I recommend to improve the results in the following way: a. Use only 422 sites with GPS data b. Use estimations of TEC, namely if error is larger than 20% of TEC the data must be discarded c. Present the results as a function of Solar Zenith Angle and Magnetic Local Time. That will be compact and informative, and there will be no necessity to plot tens of filled contours. I think that all simulated data have been stored thus it won't take a lot of time to reduce and remap

the results.

Please also note the supplement to this comment:
https://www.ann-geophys-discuss.net/angeo-2019-44/angeo-2019-44-RC2-supplement.pdf

---

## Author Comment (AC1) · 10 Oct 2019

We thank the two reviewers for their time and contributions towards the improvement of the manuscript. We have opted to respond to their comments and suggestion in a single response. Part 1 of the response deals with comments and suggestions of anonymous Reviewer #1 and Part 2 addresses those of Reviewer #2.

Part 1: Reviewer #1 Reviewer comments The paper evaluated IRI-2016 model using GPS-TEC observations during the solar minima 2008 and maxima 2013. I understand, the authors have spent a lot of time and effort on model evaluation and paper writing. However, this paper is not organized in a logic way. The authors listed many figures

in their paper but I find it difficult to understand which aspect of IRI2016 (for example: the performance in the EIA/the performance in hemispheric symmetry) they wanted to validate even though I have read their figures and the corresponding statements. I suggest the authors to rewrite it and emphasize the aspect you want to evaluate. Besides, I have some comments that need addressed in your revised version.

Response The paper evaluated the performance of IRI-2016 in reproducing TEC observed by IGS GPS observations globally using various statistical metrics. The common statistical metrics for continuous variables and categorical statistical metrics are employed. The later is intended to assess the skill of IRI-2016 at the extreme ends of TEC distribution, which is often neglected, based on quantile-based TEC categories. The scientific community has neglected this aspect primarily due to the fact that IRI model is climatological model and is not expected to reproduce extreme tails of the TEC distribution. However, in the paper, we want to investigate how good is this assumption in view of the continued improvement of IRI models over years. Therefore, we have employed categorical statistical measures to assess the skill of IRI-2016 in reproducing TEC observations at the margins of the observed distribution. In this process, we have observed a number of features from the analysis. The peculiar model skill over the EIA region and the hemispheric asymmetry in the model skill are a few of the features that we found and worth noting. Therefore, the two aspects mentioned by the reviewer are just features that came out as a result of the analysis, not the main focus on their own. In short, the study in this paper addresses the over all skill of the model across the range of TEC distribution. However, we have noted the reviewer concern and have rewritten the manuscript to improve the clarity including the aspect mentioned by the reviewer.

Major Reviewer comment 1. You used the TEC data extracted at a grid resolution of 5 latitude by 5 longitude from IGS. What is your time resolution? What are the error distributions (spatial-temporal) of IGS TEC? How about the system error? I don't think the conclusions are convinced if the errors of the GPS-TEC are not clear.

Response The time resolution of IGS TEC is two hours. The IGS TEC has estimates of TEC error at each grid. In the revised version of the manuscript, the estimated error in TEC is included and assessment is made to determine whether the difference between model and IGS TECs are within the 2-$\sigma$ error margins of IGS TEC or beyond.

Reviewer comment 2. The ability to simulate the diurnal variation of TEC is very important for a model. I suggest the authors to display some result about this performance of IRI2016.

Response This is well taken and the manuscript now includes monthly averaged diurnal model and IGS TEC at selected longitude sectors.

Reviewer comment 3. Page 10, Section 3.1.2. You display some figures at 4 selected longitudes. But you didn't say anything about the difference between different longitudes. This may be an interesting work to do. 4. Page 8, Line7-11: I think it is not appropriate to conclude that the performance during high solar activity is poor just according to the large RMSE. As we know, the TEC is larger under high solar activity than under low solar activity. When the RMSE is analyzed, the background value should not be ignored. The same situation should be considered when the RMSEs of different seasons and longitudes are analysed.

Response We agree with the reviewer that RMSE is sensitive to large model errors as compared to metrics such as mean absolute error (MAE). This may be harmful in the presence of outliers in the data sets. In particular, when the size of the data set is small, the use of MAE is preferred. Therefore, we have included MAE and skill score based on MAE. The skill score is evaluated with respect to reference measurements. In this case, we used long term climatological mean as a reference observations. In this manner, we hope to remove possible artifacts that may arise due to difference in solar activity, seasons and longitudes as indicated by the reviewer.

Reviewer comment 5. The authors should include a brief summary of the comparison between their conclusions and that of other recent publications in this field such as

"Liu, Z. et al. (2019)" and "Acharya, R., & Majumdar, S. (2019)". Additionally, a literature review of the recent publications. Liu, Z., Fang, H., Weng, L., Wang, S., Niu, J., & Meng,X. (2019). A comparison of ionosonde measured foF2 and IRI-2016 predictions over China. Advances in Space Research, 63(6), 1926-1936. Acharya, R., & Majumdar,S. (2019). Comparison of observed ionospheric vertical TEC over the sea in Indian region with IRI-2016 model. Advances in Space Research, 63(6), 1892-1904.

Response We accept the suggestion and have incorporated them in the relevant section of the revised manuscript. Details will be submitted along with revision.

Reviewer comment Minor 1. You used "S" and "O" to stand for simulated and observed data in Section 2.2.1. In Section 2.2.2, you used "SIM" and "OBS" for them, please confirm this and make them in accordance. 2. Page 7, Line 16: QPOD, not "POD". 3. Figure 4, the total number of points should be indicated. I usually do not like this kind of scatter plots that does not allow to appreciate the distribution of data. I prefer figures where the data density is more evident (e.g. using a color scale and binning the data in ranges).

Response We have color coded the latitudinal bands as suggested by the reviewer in the revised manuscript.

Part2: Reviewer #2 Reviewer comment The article considers the real problem, namely the validity of the IRI-2016 ionosphere model, which has become the most popular model up to date. The subject is suitable for the Annales Geophysicae. However, I recommend the major revision and resubmission in order to improve the results. The following shortcomings were found: 1. In the abstract and in the first chapter the authors stated the requirement of the ionosphere for the correct forecast of the radio wave propagation. However, the altitude range covered by IRI is suitable only for HF-UHF forecast. Moreover, TEC allows to predict only integral attenuation, as far as no profile properties can be derived from TEC.

Response The statements made in the abstract and the introduction are intended as

motivation and are not results of this study. It is true that TEC is a measure of integral attenuation but it is also a basis from which profile information can also be derived (e.g., using tomography techniques from slant TEC). Any improvement in understanding of TEC is critical for obtaining accurate profile informations. Therefore, we are afraid that we did not understand the shortcoming suggested by the reviewer. But we have reworded the statements for better clarity.

Reviewer comment 2. The motivation for the article is not clearly stated. IRI has been verified intensively for years, using ionozondes, satellites and GPS receivers as well. What is the novelty of the work? What is the hypothesis to be checked using GPS observations and IRI calculation? Please state it clearly!

Response We believe that we have emphasized the gap in the past studies (see also the first response to the first reviewer's comment) and how we plan to address it. Moreover, we have made some reorganization in the introduction to make this aspect notable.

Reviewer comment 3. When the authors deal with 5x5 gridded TEC values they do not work with evidence. Instead they work with the results of an IGS computer model (some kind of Kalman filter and gridding technique). Thus, the title becomes wrong – you compare one model with another model. If they want to validate IRI model then exactly 422 sites must be used, with further gridding and mapping if necessary.

Response We are afraid that the conclusion made by the reviewer is rather simplistic in reducing gridded observations to a model simulations. It is obvious that gridded observations will have additional errors due to gridding techniques on top of the observational errors (both random and sytematic (e.g., receiver bias error)). Therefore, such equivalence is inaccurate. In addition, the reviewer is suggesting to us to do our own gridding to create maps if necessary. We do not see any good reasons to do our own gridding when well investigated and quality checked gridded TEC along with associated uncertainty are readily available. Moreover, the IGS TEC is publically

accessible and the authors are not aware of publically accessible individual GPS station TEC for the whole globe except over USA. Therefore, in our opinion, comparing IRI-2016 against commonly used IGS TEC is justifiable. However, out of curiosity and respect of the reviewer suggestion, we have taken good time to derive TEC from about 200 GPS receiver sites globally and made comparison with IRI-2016 simulations. The results are presented in the form of average within latitude bands (see Fig. 1). The results have been incorporated and consistence between gridded TEC and individual GPS site TEC is indicated in the revised manuscript.

Reviewer comment 4. It is not clear, whether the authors used IRITEC subroutine, or they calculated vertical electron profiles and integrated them manually.

Response IRI has several options of providing outputs. We opted for TEC output over manual integration. We have indicated this in the revised manuscript.

Reviewer comment 5. The monthly basis can suffer from biases. It is obvious to use 27 days periods corresponding to Bartels rotation cycles. 6. Short remark about (4). The correlation coefficient (4) makes sense only for stationary processes and for the processes that have normal distribution. No tests are presented that prove the aforementioned requirements. If they are violated then the results have no sense.

Response This is indicated in the revised version.

Reviewer comment 7. Figures 2 and 3 in Mercator projection are awful and unreadable. I see the authors want to prove that they have calculated everything declared. But at that scale it is impossible to make difference between Canada and the US. It is much better to choose a couple of the most interesting frames and print them at large scale. For high latitudes the orthogonal polar projection must be chosen.

Response We have improved the readability of the figures by reducing the number of figures to allow for large scale figures.

Reviewer comment 8. The style of the presentation in the article can be accepted only

if the authors used real F10.7 and Kp indices (or IG index) from the database. But even in this case I recommend to improve the results in the following way: a. Use only 422 sites with GPS data b. Use estimations of TEC, namely if error is larger than 20% of TEC the data must be discarded c. Present the results as a function of Solar Zenith Angle and Magnetic Local Time. That will be compact and informative, and there will be no necessity to plot tens of filled contours. I think that all simulated data have been stored thus it won't take a lot of time to reduce and remap the results.

Response As indicated earlier in the response to one of the reviewer's comments, we have included GPS site based analysis to strengthen results obtained from comparison of IRI-2016 and gridded GPS TEC (e.g., see Fig. 1). We do not want to abandon the gridded IGS TEC altogether since it is widely used standard data sets in several studies. Moreover, as indicated by the reviewer, the individual IGS site TEC may have large error far exceeding the 20% of TEC in some cases. As a result, screening of bad data should be done and it is subjective in most cases (e.g., the 20% threshold suggested by the reviewer is rather rule of thumb) which suggests that model validation should be performed with standard data sets. In view of these shortcomings, we preferred to complement our studies with station-based analysis.

Summary We have noted that the comments from the reviewers are useful and helpful to improve the manuscript. As a result, we have made major revision that includes processing of GPS observables to determine TEC at about 200 stations for solar minimum 2008 and maximum 2013 for March, June, September and December months. The new processed data are then used in the comparison along with IGS gridded TEC. The details of the changes we have made in the response to the reviewers' comments are incorporated in a different letter that accompanies the revised manuscript.

[Figure]

[Figure]

**Fig. 1.** Correlation, RMSE and bias of TEC from IRI-2016 with respect to TEC drived from over 200 GPS receiver sites.

---

## Author Response (AR1)

*We thank the two reviewers for their time and contributions towards the improvement of the manuscript. We would like to note the following before directly addressing the concerns raised and suggestions made by the two reviewers. Following the feedbacks from the two reviewers, we have made major revision that includes: (I) use of additional TEC data at GPS Receiver station level in addition to the previous gridded IGS GPS-TEC; (II) new performance measures such as NMBF and NMAEF that overcomes limitation of traditional metrics when applied to data sets with different order of background values; (III) restructuring the manuscript that includes addition of new sections and removal of some sections. As a result, it is rather difficult to use track changes to indicate changes made during this revision. However, we have indicated in detail how we have addressed all the points raised by the reviewers in this response letter.*

*Moreover, in view of the new data used, associated substantial work and corresponding contributions of the authors, the order of author list is changed. The title of the manuscript is also modified to reflect the significant changes made during the revision.*

**Part 1: Reviewer #1**

**Reviewer comments**

The paper evaluated IRI-2016 model using GPS-TEC observations during the solar minima 2008 and maxima 2013. I understand, the authors have spent a lot of time and effort on model evaluation and paper writing. However, this paper is not organized in a logic way. The authors listed many figures in their paper but I find it difficult to understand which aspect of IRI2016 (for example: the performance in the EIA/the performance in hemispheric symmetry) they wanted to validate even though I have read their figures and the corresponding statements. I suggest the authors to rewrite it and emphasize the aspect you want to evaluate. Besides, I have some comments that need addressed in your revised version.

**Response**

The paper evaluated the performance of IRI-2016 in reproducing TEC observed by IGS GPS observations globally using various statistical metrics. The common statistical metrics for continuous variables and categorical statistical metrics are employed. The later is intended to assess the skill of IRI-2016 at the extreme ends of TEC distribution, which is often neglected, based on quantile-based TEC categories. The scientific community has neglected this aspect primarily due to the fact that IRI model is climatological model and is not expected to reproduce extreme tails of the TEC distribution. However, in the paper, we want to investigate how good is this assumption in view of the continued improvement of IRI models over years. Therefore, we have employed categorical statistical measures to assess the skill of IRI-2016 in reproducing TEC observations at the margins of the observed distribution. Therefore, the two aspects mentioned by the reviewer are just features that came out as a result of the analysis, not the main focus on their own. In short, the study in this paper addresses the over all skill of the model across the range of TEC distribution. However, we have noted the reviewer concern and have rewritten the manuscript to improve the clarity including the aspect mentioned by the reviewer.

**Major**

**Reviewer comment**

1. You used the TEC data extracted at a grid resolution of 5 latitude by 5 longitude from IGS. What is your time resolution? What are the error distributions (spatial-temporal) of IGS TEC? How about the system error? I don't think the conclusions are convinced if the errors of the GPS-TEC are not clear.

**Response**

The time resolution of IGS TEC is two hours. The IGS TEC has estimates of TEC error at each grid. In the revised version of the manuscript, the estimated error in TEC is included and assessment is made to determine whether the difference between model and IGS TECs are within the 2-σ error margins of IGS TEC or beyond (see page last Paragraph, page 11).

**Reviewer comment**

2. The ability to simulate the diurnal variation of TEC is very important for a model. I suggest the authors to display some result about this performance of IRI2016.

**Response**

We have checked that the diurnal cycle is well captured as it is reflected in the correlation plot which is based on a time series of TEC at 2-hour intervals. Therefore, for brevity, we have not included any figure on diurnal cycle. However, we have shown the difference in model performance during morning and afternoon hours as shown Section 3.1.2 on pages 12-13.

**Reviewer comment**

3. Page 10, Section 3.1.2. You display some figures at 4 selected longitudes. But you didn't say anything about the difference between different longitudes. This may be an interesting work to do. 4. Page 8, Line7-11: I think it is not appropriate to conclude that the performance during high solar activity is poor just according to the large RMSE. As we know, the TEC is larger under high solar activity than under low solar activity. When the RMSE is analyzed, the background value should not be ignored. The same situation should be considered when the RMSEs of different seasons and longitudes are analysed.

**Response**

We agree with the reviewer that RMSE is sensitive to large model errors as compared to metrics such as mean absolute error (MAE). This may be harmful in the presence of outliers in the data sets. In particular, when the size of the data set is small, the use of MAE is preferred. Therefore, we have included MAE and skill score based on MAE. The skill score is evaluated with respect to reference measurements. In this case, we used long term climatological mean as a reference observations. In this manner, we hope to remove possible artifacts that may arise due to difference in solar activity, seasons and longitudes as indicated by the reviewer. We have also investigated the possible impact of difference in background TEC during the two seasons on the values of the traditional model evaluation metrics such as RMSE etc. We have included recently developed metrics that overcome the limitation of RMSE, Bias and MAE when there is issues related with scale. These metrics are normalized mean bias factor and normalized mean absolute error factor. As a result, our interpretation of the model skill has changed substantially as seen in the revised manuscript throughout all the sections.

**Reviewer comment**

5. The authors should include a brief summary of the comparison between their conclusions and that of other recent publications in this field such as "Liu, Z. et al. (2019)"

and "Acharya, R., & Majumdar, S. (2019)". Additionally, a literature review of the recent publications. Liu, Z., Fang, H., Weng, L., Wang, S., Niu, J., & Meng, X. (2019). A comparison of ionosonde measured foF2 and IRI-2016 predictions over China. Advances in Space Research, 63(6), 1926-1936. Acharya, R., & Majumdar,S. (2019). Comparison of observed ionospheric vertical TEC over the sea in Indian region with IRI-2016 model. Advances in Space Research, 63(6), 1892-1904.

**Response**

We accept the suggestion and have incorporated them in the introduction section of the revised manuscript.

**Reviewer comment**

Minor 1. You used "S" and "O" to stand for simulated and observed data in Section 2.2.1. In Section 2.2.2, you used "SIM" and "OBS" for them, please confirm this and make them in accordance. 2. Page 7, Line 16: QPOD, not "POD". 3. Figure 4, the total number of points should be indicated. I usually do not like this kind of scatter plots that does not allow to appreciate the distribution of data. I prefer figures where the data density is more evident (e.g. using a color scale and binning the data in ranges).

**Response**

We have changed the plot to include only the areas between EIA crest since this region has high RMSE. Moreover, we have color coded the different seasons and separated into morning and afternoon hours as shown in Figs . 5-6 in the revised manuscript.

**Part 2: Reviewer #2**

**Reviewer comment**

The article considers the real problem, namely the validity of the IRI-2016 ionosphere model, which has become the most popular model up to date. The subject is suitable for the Annales Geophysicae. However, I recommend the major revision and resubmission in order to improve the results. The following shortcomings were found:

1. In the abstract and in the first chapter the authors stated the requirement of the ionosphere for the correct forecast of the radio wave propagation. However, the altitude range covered by IRI is suitable only for HF-UHF forecast. Moreover, TEC allows to predict only integral attenuation, as far as no profile properties can be derived from TEC.

**Response**

The statements made in the abstract and the introduction are intended as motivation and are not results of this study. It is true that TEC is a measure of integral attenuation but it is also a basis from which profile information can also be derived (e.g., using tomography techniques from slant TEC). Any improvement in understanding of TEC is critical for obtaining accurate profile informations. Therefore, we are afraid that we did not understand the shortcoming suggested by the reviewer. But we have reworded the statements for better clarity.

**Reviewer comment**

2. The motivation for the article is not clearly stated. IRI has been verified intensively for years, using ionosondes, satellites and GPS receivers as well. What is the novelty of the work? What is the hypothesis to be checked using GPS observations and IRI calculation? Please state it clearly!

**Response**

We believe that we have emphasized the gap in the past studies (see also the first response to the first reviewer's comment) and how we plan to address it. Moreover, we have made some reorganization in the introduction to make this aspect notable. We have also noted that comparison using only traditional metrics such as RMSE, bias etc has shortcoming and may lead to erroneous conclusion. This aspect has been emphasized in the revised manuscript whereby improved interpretation of the model skill is made..

**Reviewer comment**

3. When the authors deal with 5x5 gridded TEC values they do not work with evidence. Instead they work with the results of an IGS computer model (some kind of Kalman filter and gridding technique). Thus, the title becomes wrong – you compare one model with another model. If they want to validate IRI model then exactly 422 sites must be used, with further gridding and mapping if necessary.

**Response**

We are afraid that the conclusion made by the reviewer is rather simplistic in reducing gridded observations to a model simulations. It is obvious that gridded observations will have additional errors due to gridding techniques on top of the observational errors (both

random and sytematic (e.g., receiver bias error)). Therefore, such equivalence is inaccurate. In addition, the reviewer is suggesting to us to do our own gridding to create maps if necessary. We do not see any good reasons to do our own gridding when well investigated and quality checked gridded TEC along with associated uncertainty are readily available. Moreover, the IGS TEC is publically accessible and the authors are not aware of publically accessible individual GPS station TEC for the whole globe except over USA. Therefore, in our opinion, comparing IRI-2016 against commonly used IGS TEC is justifiable. However, out of curiosity and respect of the reviewer suggestion, we have taken good time to derive TEC from about 200 GPS receiver sites globally and made comparison with IRI-2016 simulations. The results related with TEC derived from indvidual GPS receiver are presented in the new Figs. 1-4, 12 .

**Reviewer comment**

4. It is not clear, whether the authors used IRITEC subroutine, or they calculated vertical electron profiles and integrated them manually.

**Response**

IRI has several options of providing outputs. We opted for TEC output over manual integration. We have indicated this in the revised manuscript.

**Reviewer comment**

5. The monthly basis can suffer from biases. It is obvious to use 27 days periods corresponding to Bartels rotation cycles.
6. Short remark about (4). The correlation coefficient (4) makes sense only for stationary processes and for the processes that have normal distribution. No tests are presented that prove the aforementioned requirements. If they are violated then the results have no sense.

**Response**

We have indicated conditions under which parametric Pearson correlation is applied. This is indicated in the revised version along with relevant literature.

**Reviewer comment**

7. Figures 2 and 3 in Mercator projection are awful and unreadable. I see the authors want to prove that they have calculated everything declared. But at that scale it is impossible to make difference between Canada and the US. It is much better to choose a couple of the most interesting frames and print them at large scale. For high latitudes the orthogonal polar projection must be chosen.

**Response**

We have improved the readability of the figures and a few of them are removed and replaced by new ones with the new GPS Receiver Station level data.

**Reviewer comment**

8. The style of the presentation in the article can be accepted only if the authors used real F10.7 and Kp indices (or IG index) from the database. But even in this case I recommend to improve the results in the following way: a. Use only 422 sites with GPS data b. Use estimations of TEC, namely if error is larger than 20% of TEC the data must be discarded c. Present the results as a function of Solar Zenith Angle and Magnetic Local Time. That will be compact and informative, and there will be no necessity to plot tens of filled contours. I think that all simulated data have been stored thus it won't take a lot of time to reduce and remap the results.

**Response**

As indicated earlier in the response to the first reviewer's comments, we have included GPS site based analysis to strengthen results obtained from comparison of IRI-2016 and gridded GPS TEC . We do not want to abandon the gridded IGS TEC altogether since it is widely used standard data sets in several studies. Moreover, as indicated by the reviewer, the individual IGS site TEC may have large error far exceeding the 20% of TEC in some cases. As a result, screening of bad data should be done and it is subjective in most cases (e.g., the 20% threshold suggested by the reviewer is rather rule of thumb) which suggests that model validation should be performed with standard data sets. In view of these shortcomings, we preferred to complement our studies with station-based analysis.

**Summary**

We have noted that the comments from the reviewers are useful and helpful to improve the manuscript. As a result, we have made major revision that includes processing of GPS observables to determine TEC at about 200 stations for solar minimum 2008 and

maximum 2013 for March, June, September and December months. The new processed data are then used in the comparison along with IGS gridded GPS-TEC.

---

## Referee Report (RR1)

In the manuscript, the authors evaluate the IRI performance comprehensively by using GPS TEC data, and the results are convincing. Moreover, the manuscript is well organized, so I think it is suitable for publication after making minor corrections.

Minor comments:

1. For the abbreviation "EIA" in the abstract, full name should be given when appearing for the first time.

2. In Section 2.2.1, when explaining why using the Pearson correlation analysis, the author mentions "Several studies have shown that TEC data for a limited number of days fulfills both of these conditions.", I think it is better to add some relevant references.

3. Page 9 line 10 and Equation (13): replace POD with QPOD?.

4. In the Results and Discussions part, the arrangement of some pictures is unreasonable, which makes it inconvenient to read. For example, Figures 2-4 are mainly mentioned in Section 3.1.1, but the author puts them at the end of Section 3.1.2. While Figures 5 and 6, mentioned in Section 3.1.2, are put in Section 3.1.3. I hope the author will review and organize them in a clear way.

---

## Editor Decision (ED1)

Dear Dr. Mulugeta Melaku,

Thank you for submitting your manuscript to Annales Geophysicae. I appreciate your efforts when preparing such extensive comparative analysis. Recently the discussion on the manuscript is closed and I am coming to you on the status of your paper.

In general, the manuscript is well-written and the results are important for ionospheric and HF communities. Nevertheless, the paper has some shortages and inaccuracies. I guess that there is important to mention in the paper that although year 2013 was in the middle of the 24[th] solar cycle, but at the same time between two peaks of the maximum (not the maximum itself). During 2013 there occurred only three strong magnetic storms with Dst<-100 during 2013 (in the middle of march and at the beginning and at the end of June. On the other hand, the whole 24[th] solar cycle is significantly lower comparing with previous ones (see the figures below). My other additional comment is that the quiet geomagnetic conditions are considered when the Dst values not less than -20nT (Dst values lower than -20 nT up to -50 nT means minor storm conditions). Also, according to both referee's comments, there is necessary to clarify some statements and information given.

[Figure]

Therefore, I suggest a major revision of the manuscript. Please, consider carefully and discuss in the revised version of the manuscript comments of both referees. One of them and me will read the manuscript again.

If you decide to revise the work, please submit a list of changes or a rebuttal against each point which is being raised when you submit the revised manuscript.

Kindest regards

Yours sincerely

D. Buresova

---

## Author Response (AR2)

May 10, 2020

**Subject: Submission of revised manuscript**

Dear Prof. Buresova,

I would like to thank you and the anonymous reviewers for their time and inputs. We have incorporated the minor suggestions from the reviewers and yourself in the manuscript as indicated in the abstract (line 13 to 17), in the body of the manuscript (page 15, lines 4 to 20; page 16, line 1 to 10 including Fig. 7), and in the conclusion (page 27, line 4-6).

I am looking forward to your positive decision.

Sincerely yours,

Gizaw Mengistu Tsidu (Prof.)